# Active foot placement control ensures stable gait: Effect of constraints on foot placement and ankle moments

**A. M. van Leeuwen**[1,2], **J. H. van Dieën**[1], **A. Daffertshofer**[1,2], **S. M. Bruijn**[1,2,3]*

**1** Department of Human Movement Sciences, Faculty of Behavioural and Movement Sciences, Vrije Universiteit Amsterdam, Amsterdam Movement Sciences, Amsterdam, The Netherlands, **2** Institute of Brain and Behavior Amsterdam, Amsterdam, The Netherlands, **3** Biomechanics Laboratory, Fujian Medical University, Quanzhou, Fujian, PR China

* s.m.bruijn@gmail.com

**Data Availability Statement:** The data and analysis can be found within Zenodo through the following link https://doi.org/10.5281/zenodo.4229851.

## Abstract

Step-by-step foot placement control, relative to the center of mass (CoM) kinematic state, is generally considered a dominant mechanism for maintenance of gait stability. By adequate (mediolateral) positioning of the center of pressure with respect to the CoM, the ground reaction force generates a moment that prevents falling. In healthy individuals, foot placement is complemented mainly by ankle moment control ensuring stability. To evaluate possible compensatory relationships between step-by-step foot placement and complementary ankle moments, we investigated the degree of (active) foot placement control during steady-state walking, and under either foot placement-, or ankle moment constraints. Thirty healthy participants walked on a treadmill, while full-body kinematics, ground reaction forces and EMG activities were recorded. As a replication of earlier findings, we first showed step-by-step foot placement is associated with preceding CoM state and hip ab-/adductor activity during steady-state walking. Tight control of foot placement appears to be important at normal walking speed because there was a limited change in the degree of foot placement control despite the presence of a foot placement constraint. At slow speed, the degree of foot placement control decreased substantially, suggesting that tight control of foot placement is less essential when walking slowly. Step-by-step foot placement control was not tightened to compensate for constrained ankle moments. Instead compensation was achieved through increases in step width and stride frequency.

## Introduction

On every step we take, our center of mass (CoM) accelerates laterally towards the new stance foot. In order not to fall, this motion of the CoM has to be reversed, preventing the CoM to move beyond the lateral border of the base of support [1]. The moment that accelerates the CoM in the opposite direction can be controlled by adjusting the center of pressure (CoP) [2]. During gait, the dominant mechanism to control the CoP is the so-called foot placement strategy [1, 3].

**Funding:** Sjoerd M. Bruijn and Moira van Leeuwen were supported by a grant from the Netherlands Organization for Scientific Research (016. Vidi.178.014), https://www.nwo.nl/en/. The funders had no role in study design, data collection and analysis, decision to publish, or preparation of the manuscript.

**Competing interests:** The authors have declared that no competing interests exist.

**Abbreviations:** CoM, Center of mass; CoP, Center of pressure; BF10, Bayes factor indicating evidence supporting the alternative hypothesis; BF01, Bayes factor indicating evidence supporting the null hypothesis; FP, The mediolateral (along the global x-axis) foot position (calcaneus position digitized with respect to the foot cluster marker) at midstance expressed with respect to the mediolateral position of the contralateral foot at midstance (i.e. step width). The variable was demeaned prior to the performed regression (model 1); $CoM_{pos}$, Mediolateral CoM position (along the global x-axis) expressed with respect to mediolateral position of the stance foot at midstance. This variable was demeaned prior to the performed regression (model 1); $\beta_{pos}$, Regression coefficient defining the relationship between mediolateral CoM position ($CoM_{pos}$) and step width (FP), as part of the foot placement model (model 1); $CoM_{vel}$, Mediolateral CoM velocity (along the global x-axis), calculated as the derivative of the CoM position expressed in the global coordinate system. This variable was demeaned prior to the performed regression (model 1); $\beta_{vel}$, Regression coefficient defining the relationship between mediolateral CoM velocity ($CoM_{vel}$) and step width (FP), as part of the foot placement model (model 1); FP2, The mediolateral (along the global x-axis) foot position (calcaneus position digitized with respect to the foot cluster marker) at midstance expressed with respect to the CoM at the time of toe-off in accordance with Rankin et al. [12]. The variable was demeaned and divided by the standard deviation prior to the performed regression (model 2); $EMG_{gm\_swing}$, The median gluteus medius EMG amplitudes over 60–80% of the gait cycle (early swing), multiplied by the duration of this episode in seconds. The variable was demeaned and divided by the standard deviation prior to the performed regression (model 2); $\beta_{gm\_swing}$, Regression coefficient defining the relationship between gluteus medius activity during early swing ($EMG_{gm\_swing}$) and step width (FP2), as part of the muscle model (model 2); $EMG_{al\_swing}$, The median adductor longus EMG amplitudes over 60–80% of the gait cycle (early swing), multiplied by the

Wang and Srinivasan [4] captured this control strategy in a linear model, indicating that foot placement can be predicted by the CoM kinematic state (i.e. CoM position and velocity) during the preceding swing phase. The association between foot placement and CoM kinematic state was shown to be less pronounced during walking with external lateral stabilization [5], which supports the notion that step-by-step foot placement control promotes mediolateral gait stability.

In healthy individuals, ankle, hip and push-off strategies complement foot placement in maintaining stability [2, 6]. Coordinated recruitment of these control strategies may guarantee stability throughout the gait cycle [7, 8]. For example, the ankle strategy allows for an early response to a perturbation, before the foot placement strategy becomes effective following heel strike [9]. In elderly, pathological, or prosthetic gait, an impairment in one strategy may require individuals to rely more on another one [10, 11]. When aiming for stability improvements, it is hence important to understand the details of these control strategies as well as their interplay. In the current study, we focused on the foot placement strategy in healthy individuals during steady-state treadmill walking. Following Wang and Srinivasan [4], we expected that mediolateral foot placement can be predicted by CoM state, reflective of a step-by-step control strategy (E1). And, in line with Rankin et al. [12], we expected mediolateral foot placement to correlate with hip ab- and adductor muscle activity during the preceding swing phase (E2), supporting the active nature of this control mechanism. We tested these two expectations during normal and slow walking. We chose two speeds because the implementation of the foot placement strategy has been shown to be speed-dependent [13]. We did not make any speed-related predictions, but the speed-dependent nature of the foot placement strategy could potentially affect the effects of our other experimental conditions. These experimental conditions served to challenge the degree of (active) foot placement control; see below.

## Degree of control

The degree of foot placement control can be inferred from its predictability based on the CoM state. The mid-swing CoM state is a better predictor of foot placement than the swing foot state itself, although this prediction is not entirely accurate [4]. Eventual inaccuracies could be attributed to motor noise, or task constraints. In certain contexts, a high degree of foot placement control might not be necessary. As an example, we note that foot placement seems to correlate less with CoM state in externally stabilized gait, and at slower speeds. Apparently, its control is less important in these conditions [5, 13]. In addition, the necessary degree of foot placement control may depend on the availability of alternative control strategies. Arguably, foot placement is most effective in shifting the CoP, though adapting ankle moments allows for a complementary shift during stance. As mentioned above, this can enable early stabilizing responses [9] and/or corrections of inaccurate foot placements [11]. According to Fettrow et al. [8], foot placement and ankle moments are interdependent, i.e. one may compensate for the other. In this context, we hypothesized foot placement to compensate for a limited possibility to shift the CoP underneath the stance foot, through constrained ankle moments (H1). If true, this will be in line with the findings of Hof et al. [11], who demonstrated that more lateral foot placement compensated for the impossibility to induce a CoP shift under a prosthetic leg. Conversely, a lesser degree of foot placement control might be compensated for by ankle moments. Such compensation could facilitate adaptation of foot placement to environmental or task constraints without threatening stability. Accordingly, we hypothesized that constraining the foot placement by stepping onto projected lines will yield diminished foot placement control (H2).

duration of this episode in seconds. The variable was demeaned and divided by the standard deviation prior to the performed regression (model 2); $\beta_{al\_swing}$, Regression coefficient defining the relationship between adductor longus activity during early swing ($EMG_{gm\_swing}$) and step width (FP2), as part of the muscle model (model 2); $R^2$, The relative explained variance of models (1) and (2). For model 1 this variable is interpreted as a measure of the degree of foot placement control. For model 2 this variable is interpreted as a measure of the active contribution to step-by-step foot placement control.

## Active control

Whether foot placement emerges mainly from a passive rather than an active mechanism is currently under debate [14], though there is increasing evidence for the latter. An illusory perturbation of the CoM state by manipulating muscle spindle afference has been reported to lead to predictable adjustments in foot placement, which indicates it to be affected by sensory input [15]. Similarly, illusory falls induced by visual or vestibular stimulation seem to trigger reactive foot placement adjustments [7, 16]. Both in response to perturbations [3] as well as during steady-state walking [12], gluteus medius activity (hip abductor muscle) has been associated with step width. In addition to replicating the aforementioned findings of Rankin et al. [12], we therefore hypothesized that constraining ankle moments will yield a larger active contribution to step-by-step variability in mediolateral foot placement, indicative of a compensatory foot placement strategy (H3).

With our expectations, E1 & E2, we aimed at replicating findings [4, 12] of actively controlled step-by-step foot placement during steady-state walking. We also sought to investigate how important this foot placement strategy is, by evaluating whether we can constrain foot placement (H2) during steady-state walking and whether foot placement control would tighten as a compensatory mechanism, when ankle moments are constrained (H1 & H3).

This study's preregistered hypotheses, protocol and sampling plan can be found on OSF: https://osf.io/74pn5.

## Methods

### Participants

Only participants capable of walking without difficulty for a longer duration ($\geq$ 60 minutes) were recruited. Participants were excluded when they reported sports injuries or other motor impairments that possibly affected their gait pattern. Participants suffering from (self-reported) balance issues were also excluded.

An initial sample size of ten participants was recruited as we used a Bayesian sequential sampling approach. Subsequently, recruitment of participants continued until a threshold of meaningful evidence was reached [17]. We set this threshold to a BF10 or BF01 of 10 (indicative of strong relative evidence) for either the null or the alternative hypothesis, based on our main outcome measures. Since not all outcome measures reached a BF10 or BF01, we continued recruitment until the pre-determined maximum of 30 participants was attained, while accounting for drop-outs.

A total of 35 healthy participants completed the experiment according to instructions. The data of four participants had to be discarded because of technical malfunctioning of equipment, and data of one participant were excluded in view of pronounced toeing out during normal walking. Finally, data of 30 participants (19 female, 11 male; 30 ± 7 yrs, 70 ± 13 kg, 1.73 ± 0.08 m; mean ± sd) were included in the analysis.

Prior to participation, participants signed an informed consent and ethical approval (VCWE-2018-159) was granted by the ethics review board of the faculty of Behavioural and Movement Sciences at 'Vrije Universiteit Amsterdam'.

### Procedure

Participants were invited to walk on a treadmill during four conditions (see Fig 1) at normal ($1.25 \times \sqrt{}$(leg length) m/s) and slow ($0.63 \times \sqrt{}$(leg length) m/s) walking speeds, normalized to leg length in accordance with Hof [18]. In this study, stride frequency control could be a confounding stability control strategy [19]. Therefore, stride frequency was controlled by means of a metronome, to avoid that participants changed stride frequency between conditions.

| Condition | Symbol | Description |
|---|---|---|
| **Steady-state walking** |  | Steady-state walking without any constraints. |
| **Ankle moment constrained** |  | Walking with shoes with a narrow ridge (1 cm) underneath the soles (LesSchuh, Fig 1), constraining mediolateral displacement of the CoP underneath the foot in a straight line. |
| **Foot placement constrained** |  | Walking with bilateral foot placement constraints (projected lines on treadmill indicating mediolateral target locations for foot placement). |

**Fig 1. Conditions performed at normal and slow walking speeds.**

Participants were instructed to time either their right or their left heel strikes to the metronome beat. The imposed frequency was set to the average preferred stride frequency determined during the final 100 steps of a familiarization trial for each speed (see below).

We randomized the order of the conditions across participants and speeds. Before beginning the experiment, participants performed a five-minutes familiarization phase (two minutes at normal walking speed, three minutes at slow walking speed) without imposing further constraints; cf. below. Additionally, participants were familiarized with ankle moment constraining shoes (Fig 2) prior to data collection. To ensure that all trials contained at least 200 consecutive strides, trials at normal walking speed lasted five minutes and trials at slow walking speed ten minutes. Between trials, sufficient breaks were provided to prevent fatigue as verified by subjective report.

## Constraints

**Foot placement constraint.** Projections on the treadmill served to constrain the variation in mediolateral foot placement. In brief, first the average step width was derived from the final 100 steps of the familiarization trial, based on the CoP estimated from the force measurement of the instrumented treadmill. Then, this average step width was imposed by projecting beams on the treadmill, and participants were instructed to place their foot in the middle of the beam. For every step, the beam became visible following toe-off in order to prevent modification of the CoM swing phase trajectory by compensatory push-off modulation [20]. Customized labview software allowed us to estimate the toe-off event based on a force threshold, which triggered the projections based on real-time force measurements.

**Ankle moment constraint.** The mediolateral shift of the CoP underneath the stance foot was minimized using a shoe with a narrow (1 cm) ridge attached to the sole. This so-called LesSchuh limits ankle moments while anteroposterior roll-off and subsequent push-off remain possible, because the material of the ridge bends with the sole in anterior-posterior direction. Participants were asked to walk on the ridge, not touching the ground with the sides of the shoe's sole. Participants were also instructed to place their feet in a similar orientation as they would without the ankle moment constraint, to avoid a "toeing-out strategy" [21] potentially inducing a mediolateral shift of the center of pressure after foot placement (heel strike) despite the narrow base of support.

## Data collection

**Force plate.** Participants walked on an instrumented dual-belt treadmill (Motek-Forcelink, Amsterdam, Netherlands). Ground reaction forces and moments were recorded from the

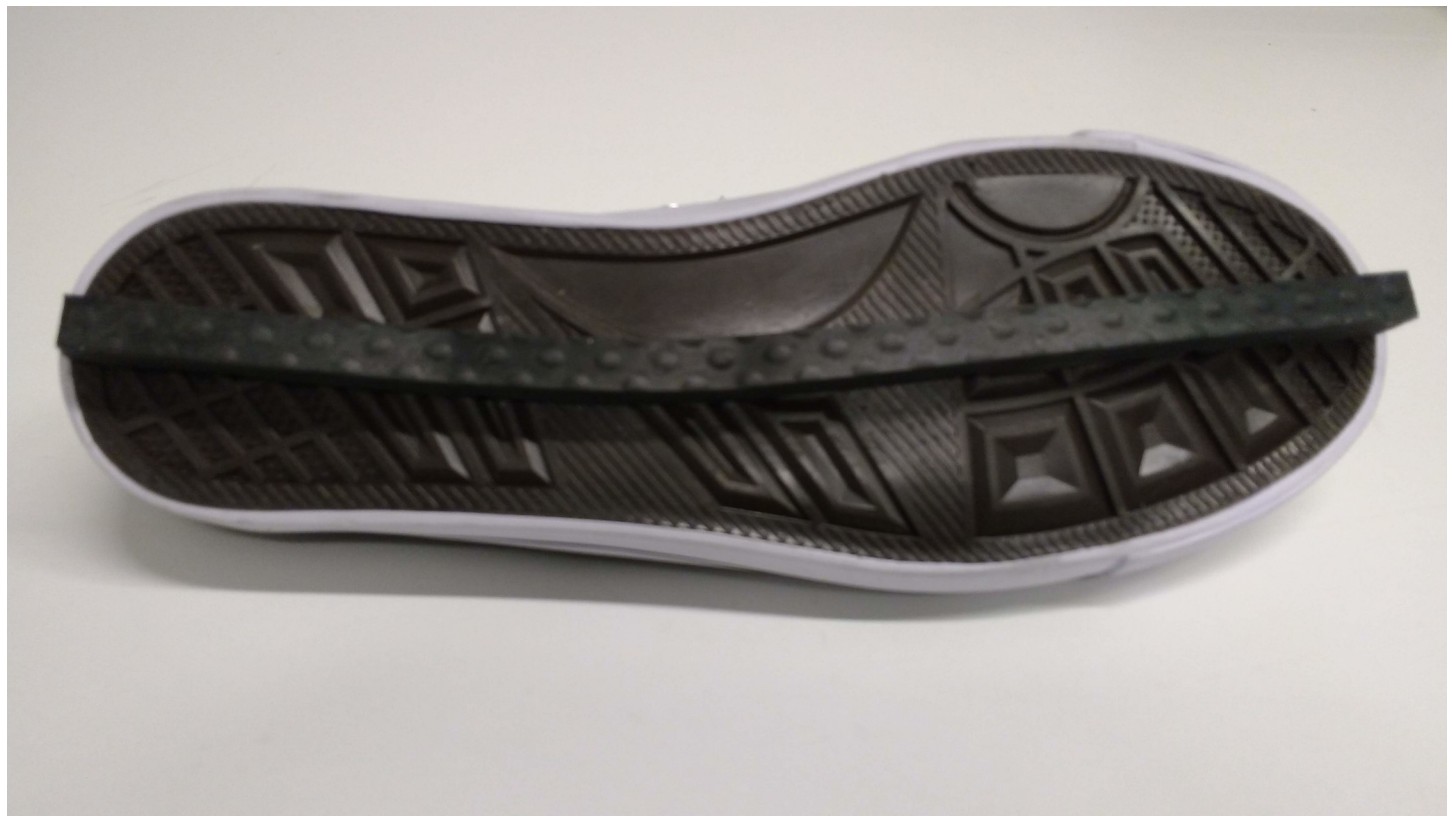

**Fig 2. LesSchuh.** Shoe with an ankle moment constraint (width of the ridge is 1 cm).

force plates embedded in the treadmill and sampled at 200 Hz. From these forces, we determined the CoP.

**Optotrak.** Full body kinematics were measured using two Optotrak cameras (Northern Digital Inc, Waterloo Ontario, Canada) directed at the center of the treadmill (sampling rate was 50 Hz). Cluster markers were attached to the feet, shanks, thighs, pelvis, trunk, upper arms and forearms. Corresponding anatomical landmarks were digitized using a six-marker probe.

**Electromyography.** Bipolar surface electromyography (EMG) was applied to measure bilateral muscle activity from 16 muscles expected to contribute to gait stability. Here, we concentrated on the m. gluteus medius and m. adductor longus muscle given their hypothesized contribution to foot placement control (for other muscles recorded, we refer to the preregistered protocol: https://osf.io/74pn5). A 16-channel Porti EMG amplifier (TMSi, Enschede, Netherlands) served to record the two muscles bilaterally; signals were sampled at 2 kHz. Surface EMG electrodes with a diameter of 22 mm were positioned on the skin according to the SENIAM guidelines [22].

**Data analysis.** For all subjects and trials, we analyzed 200 consecutive strides. These were the final 200 strides of every trial, unless data quality urged selection of earlier strides (e.g., better marker visibility, less noise).

## Data processing

**Gait event detection.** Gait events (heel strikes & toe-offs) were detected based on the characteristic "butterfly pattern" of the combined center of pressure as derived from force plate data [23]. We defined a step as the period between toe-off and heel strike. Mid-swing was defined at 50 percent of the step.

**Center of mass.**  For every segment, we estimated the segment's mass via linear regression including the segment's length and the segment's measured circumference as predictors and regression coefficients based on gender [24]. The segment's CoM was estimated as a percentage of the longitudinal axis of the segment [24, 25]. The full body CoM was derived from a weighted sum of the body segment's CoMs. Mediolateral CoM displacement was defined along the x-axis of our global coordinate system, of which the x-axis was oriented perpendicular to the direction of the treadmill. We determined the increment of the mediolateral CoM position $CoM_{pos}$, divided by the time-step, as estimate of the mediolateral CoM velocity $CoM_{vel}$.

**EMG processing.**  EMG data were high-pass filtered at 20 Hz, rectified, and low-pass filtered at 50 Hz, following Rankin et al. [12]. Strides were time-normalized to 1000 samples. For every stride, we determined $EMG_{gm\_swing}$ and $EMG_{al\_swing}$ as the median EMG activity during early swing (60–80% of the stride cycle) multiplied by the duration of this episode in seconds (we note here that this is a deviation from our preregistered protocol, see "Deviations from the preregistered plans I").

## Outcome measures

**Multiple linear regression models.**  We used a multiple linear regression with mediolateral foot placement (FP, i.e. the mediolateral foot position at midstance expressed with respect to the mediolateral position of the contralateral foot at midstance (step width), as dependent variable and the CoM state variables ($CoM_{pos}$, $CoM_{vel}$), i.e. the CoM position expressed with respect to the position of the stance foot at midstance and CoM velocity, as independent variables (i.e. predictors). The predictors' time series were time normalized to 51 samples for every step, from toe-off to heel strike. FP, $CoM_{pos}$ and $CoM_{vel}$ were demeaned prior to regression.

In a separate model, mediolateral foot placement (FP2), i.e. the mediolateral foot position at midstance with respect to the CoM at the time of toe-off, was the dependent variable with the median gluteus medius and adductor longus EMG amplitudes from 60–80% of the gait cycle ($EMG_{gm\_swing}$, $EMG_{al\_swing}$) as independent variables. In this model, within subjects and trials, we normalized FP2, $EMG_{gm\_swing}$ and $EMG_{al\_swing}$ by demeaning and dividing by the standard deviation (we note here that this is a deviation from our preregistered protocol, see "Deviations from the preregistered plans II").

The first model was created following Wang and Srinivasan [4] and read:

$$FP = \beta_{pos} \cdot CoM_{pos}(i) + \beta_{vel} \cdot CoM_{vel}(i) + \varepsilon_1(i), \tag{1}$$

in which $\varepsilon_1$ denotes the error in every step cycle, for different phases (i) of the step cycle. Statistical tests were performed for CoM state predictors at mid-swing (mid, i = 25) and terminal swing (ts, i = 51).

The second model had the form:

$$FP2 = \beta_{gm_s wing} \cdot EMG_{gm_s wing} + \beta_{al_s wing} \cdot EMG_{al_s wing} + \varepsilon_2. \tag{2}$$

We quantified the degree of foot placement control via the relative explained variance ($R^2$) of model 1 and the active contribution to step-by-step variability in foot placement via the $R^2$ of model 2.

The data and analysis can be found on Zenodo: 10.5281/zenodo.4229851.

## Statistics

All statistical tests were performed in JASP (JASP Team (2019). JASP (Version 0.11.1)).

**Step-by-step foot placement control in steady-state walking.**  The regression coefficients (β) of model (1) were tested against zero in a Bayesian one-sample t-test, to infer whether mediolateral foot placement can be predicted by CoM state during steady-state walking (E1) (we note here that this is a deviation from our preregistered protocol, see "Deviations from the preregistered plans III"). And, to assess whether mediolateral foot placement correlates with hip ab- and adductor muscle activity of the preceding swing phase during steady-state walking (E2), we tested the regression coefficients of model (2) against zero in a Bayesian one-sample t-test.

**Step-by-step foot placement control with ankle moment constraints.**  We performed a Fisher transformation on the $R^2$ values prior to statistical testing. Bayesian equivalents of a 2×2 repeated measures ANOVA with factors *Condition* (levels: ankle moment constrained/foot placement constrained versus steady-state walking) and *Speed* (levels: normal versus slow) served to test the effects of the constraints and walking speed, as well as their interaction, on degree of (active) step-by-step foot placement control; relative explained variance of models (1) and (2). Moreover, to test the constrained conditions against the steady-state walking condition, we used Bayesian planned post-hoc assessments.

We tested the $R^2$ values of models (1) and (2) in the ankle moment constrained condition against the steady-state walking condition. By this we could estimate whether constraining the ankle moment led to compensation in the degree of foot placement control. That is, this allowed for testing the hypotheses that compensation will encompass tighter control (i.e. a higher $R^2$ of model (1)) (H1) driven by compensatory muscle activation (i.e. a higher $R^2$ of model (2)) (H3). If both H1 and H3 are true, this would suggest a compensatory tighter coupling between variations in foot placement and CoM state (H1), achieved through compensatory muscle activity (H3). If only H1 is true, this would suggest tighter coupling between variations in foot placement and CoM state (H1), but no evidence that this is actively controlled (in contrast with H3). If only H3 is true, this would suggest that foot placement is more strictly controlled through muscle activity (H3), but not to tighten the coupling between variations in foot placement and CoM state.

**Step-by-step foot placement control with foot placement constraints.**  In order to infer whether the degree of control decreased (i.e. a lower $R^2$ of model (1)) (H2), when constraining foot placement, we tested the $R^2$ of model (1) in the foot placement constrained condition against the steady-state walking condition.

**Deviations from the preregistered plans.**  I) In our preregisterd plans (https://osf.io/74pn5), we intended to take the integral of the EMG signal over the selected time period. To avoid amplification of artefacts by taking the integral, while still retaining the influence of time, we deviated from the preregistered plans and computed the product of the median EMG amplitude and time in seconds.

II) The definition of mediolateral foot placement (FP2) and normalization of FP2, $EMG_{gm\_swing}$ and $EMG_{al\_swing}$ by both subtracting the mean and dividing by the standard deviation was not included in detail in the preregistered plans, but is in line with Rankin et al. [12].

III) In our preregistration we planned to bootstrap the regression coefficients for each participant. However, although most participants, but not all, demonstrated significant relationships, the final conclusion was based on statistics on group level and included in the results.

## Results

### Step-by-step foot placement control in steady-state walking

**Model 1: Foot placement model.**  First, we considered the relative explained variance ($R^2$) and tested the regression coefficients of the foot placement model (1) to assess our expectation

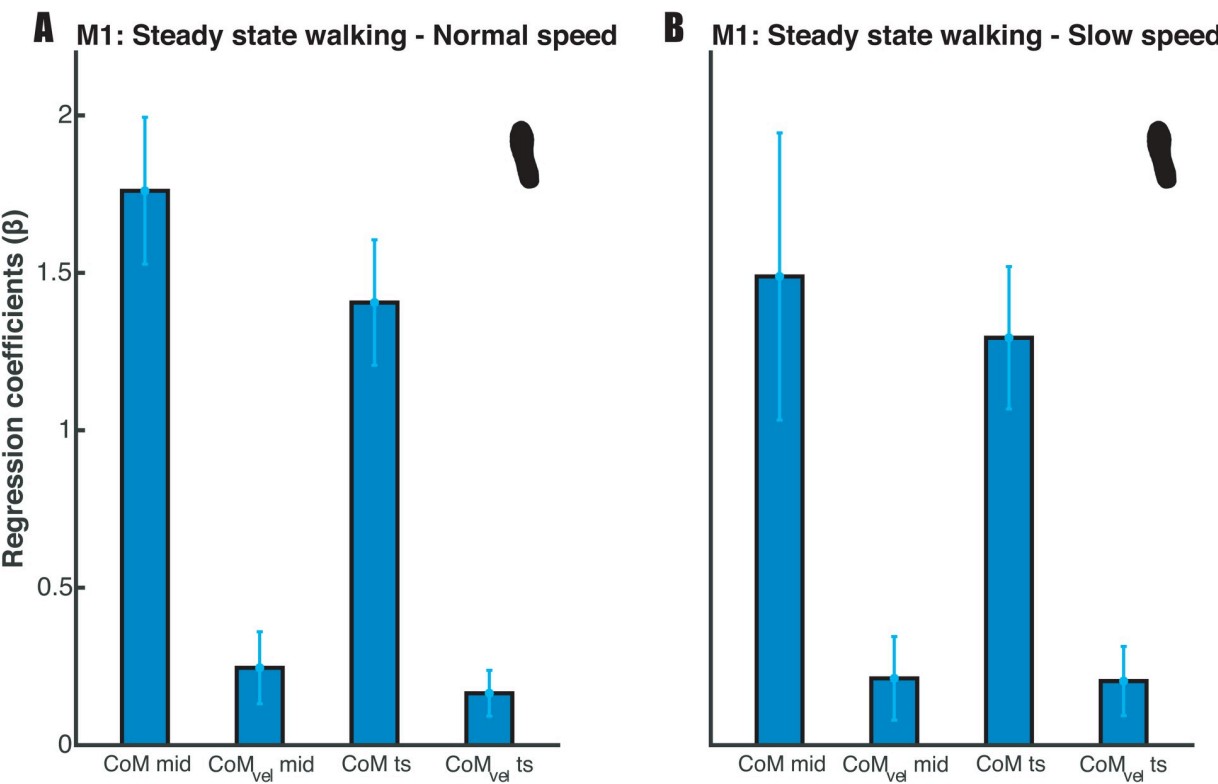

**Fig 3. Mean regression coefficients of the foot placement model (1).** Standard deviations are represented by error bars. Panels A and B represent the results for respectively normal and slow walking speed. The beta coefficients were tested at mid-swing and terminal swing, demonstrating extreme evidence ($BF_{10} > 100$) for inclusion of the predictors.

(E1) that mediolateral foot placement can be predicted by CoM state, as previously shown by Wang and Srinivasan [4]. In the steady-state walking condition (unconstrained), the foot placement model predicted over 60% of the variance at normal walking speed and over 40% at slow walking speed. There was extreme evidence ($BF_{10} > 100$), when testing the regression coefficients for $CoM_{pos}$ (at mid-swing & terminal swing) and $CoM_{vel}$ (at mid-swing and terminal swing) against zero (Fig 3), that swing phase CoM state predicts foot placement during steady-state walking (E1).

**Model 2: Muscle model.** Our second expectation (E2), that mediolateral foot placement correlates with hip ab- and adductor muscle activity during the preceding swing phase, was tested based on the regression coefficients of the muscle model (2), and visualized in Figs 4 and 5 below. Fig 4 illustrates a similar result as in Rankin et al. [12] for the relation between foot placement and gluteus medius (hip abductor) activity. More lateral steps were associated with higher bursts in gluteus medius activity during early swing (60–80% of the stride cycle). Conversely, as shown in Fig 5, higher adductor longus activity was associated with more medial steps.

Our results show that, in the steady-state walking condition, m. gluteus medius and m. adductor longus activity predicted foot placement. At normal walking speed, we found extreme ($BF_{10\_gm} = 50281.309$) and moderate evidence ($BF_{10\_al} = 6.050$) when testing the muscle model's (model 2) regression coefficients against zero (Fig 6A). At slow walking speed, extreme evidence was found for both muscles' regression coefficients ($BF_{10\_gm} = 535.867$, $BF_{10\_al} = 4984.586$, Fig 6B). The sign of the regression coefficients was as expected, with more gluteus medius activity corresponding to more lateral foot placement (positive sign) and more

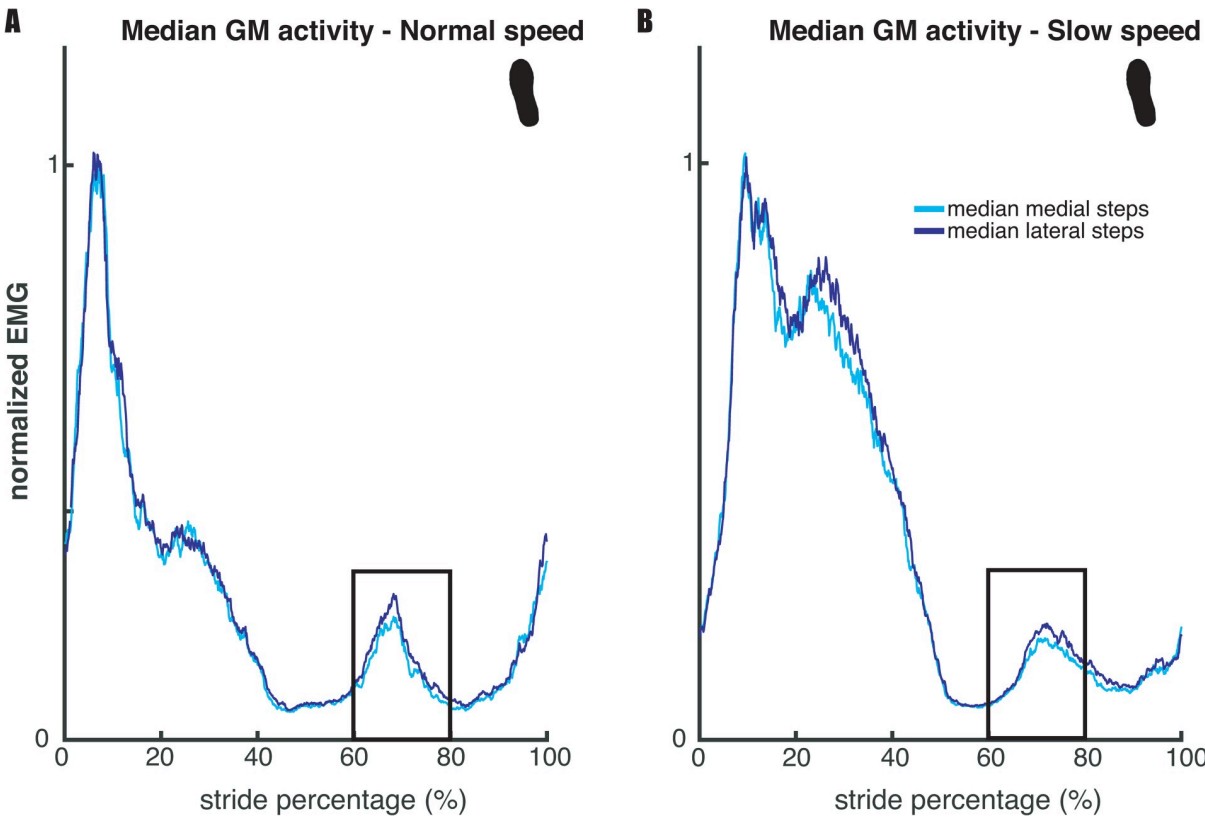

**Fig 4. Median gluteus medius activity across legs and participants.** Panels A and B show the results for respectively normal and slow walking speed. For each participant strides were divided over medial and lateral steps, of which the median was taken respectively. For the median lateral step, there was a higher burst in gluteus medius activity during early swing (60–80%) of the gait cycle. The depicted EMG traces are normalized to average stride peak activity for each speed respectively. The figure serves as a dichotomous illustration of the relationship established through regression and does not show values that were statistically tested.

medial foot placement corresponding to more adductor longus activity (negative sign). Although the mean $R^2$ was low for both speeds ($M_{normal\_speed} = 0.0268$; $M_{slow\_speed} = 0.0258$) see also Fig 8, below), the moderate to extreme evidence for the regression coefficients supports the idea that mediolateral foot placement is determined by hip ab- and adductor muscle activity during the preceding swing phase (E2).

## Step-by-step foot placement control with ankle moment constraints

**Model 1: Foot placement model.** Regarding our first hypothesis (H1), that constrained ankle moments would lead to tighter foot placement control, we tested the relative explained variance ($R^2$) of the foot placement model (1). The Bayesian repeated measures ANOVA of $R^2$ of the foot placement model (model 1, Fig 7) revealed that the best model included only the factor *speed*. There was extreme evidence for this model as compared to the Null model as tested at mid- and terminal swing ($BF_{10\_mid} = 4.732 \cdot 10^{15}$, $BF_{10\_ts} = 2.233 \cdot 10^9$). Since the inclusion of the factor *condition* did not improve the model, we could not find evidence for an effect of condition on the $R^2$. Taken together, foot placement did not compensate for the constrained ankle moments by more accurate control (in contrast with H1).

**Model 2: Muscle model.** Regarding our third hypothesis (H3), that constrained ankle moments would lead to larger active contribution to step-by-step variability in mediolateral foot placement, we tested the relative explained variance ($R^2$) of the muscle model (2). The

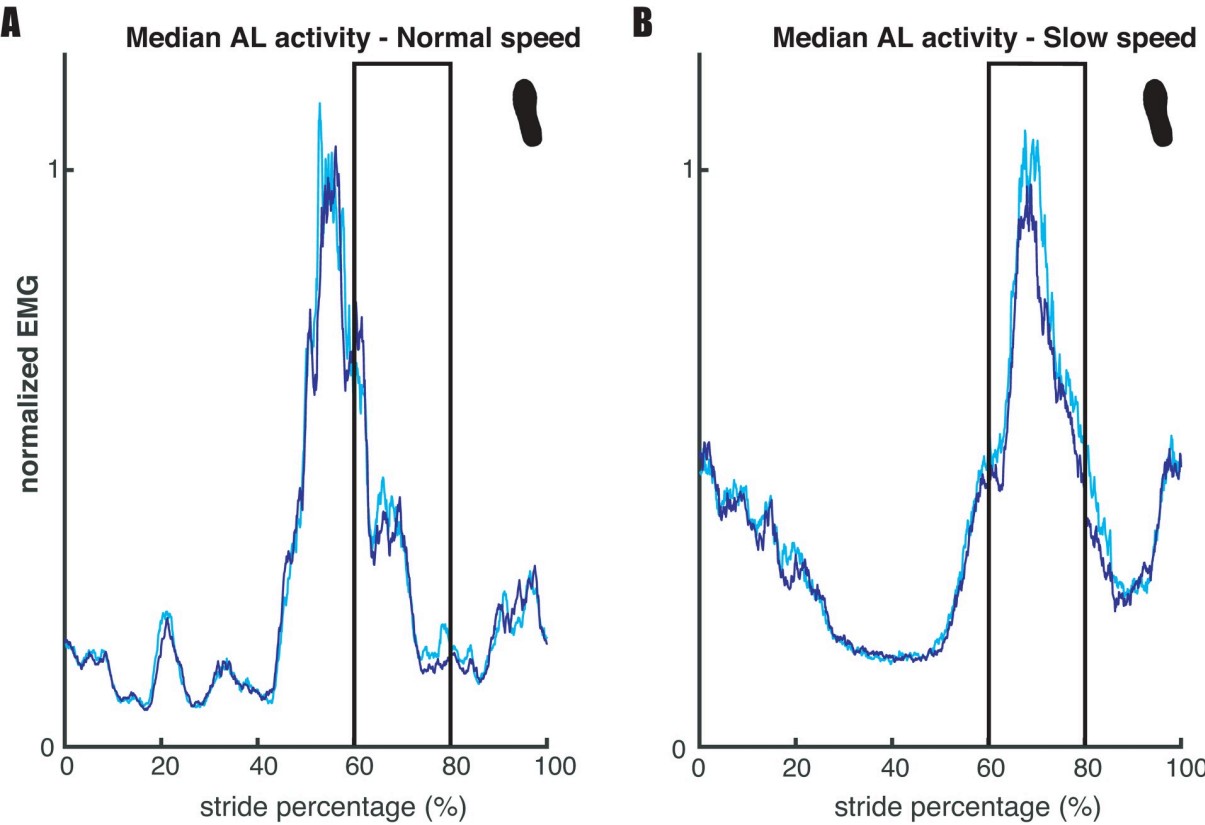

**Fig 5. Median adductor longus activity across legs and participants.** Panels A and B represent the results for respectively normal and slow walking speed. For each participant strides were divided over medial and lateral steps, of which the median was taken respectively. When comparing medial to lateral steps during early swing (60–80% of the gait cycle), higher EMG activity appears to be associated with more medial steps. This is more prominent at slow walking speed. The depicted EMG traces are normalized to average stride peak activity for each speed, respectively. The figure serves as a dichotomous illustration of the relationship established through regression and does not show values that were statistically tested.

Bayesian repeated measures ANOVA of $R^2$ of the muscle model (model 2, Fig 8) revealed that the best model included only the factor *condition*. There was anecdotal evidence for this model as compared to the Null model ($BF_{10}$ = 1.786). Post-hoc analysis also provided anecdotal evidence ($BF_{10}$ = 1.374) for compensatory muscle activity (H3) in the ankle moment constrained condition as compared to steady-state walking. As such, the effect of the ankle moment constraint on the relationship between muscle activity and foot placement remains inconclusive.

## Step-by-step foot placement control with foot placement constraints

**Model 1: Foot placement model.** Regarding our second hypothesis (H2), that the degree of foot placement control would decrease, when constraining foot placement, we tested the relative explained variance ($R^2$) of the foot placement model (1). The Bayesian repeated measures ANOVA of the $R^2$ of the foot placement model (model 1, Fig 9) revealed that the best model included the factors *condition* and *speed*, when testing for mid-swing. There was extreme evidence for this model as compared to the Null model ($BF_{10\_mid}$ = 3.196·$10^{23}$). Post-hoc analysis provided strong evidence ($BF_{10\_mid}$ = 19.125) in favor of a poorer prediction of foot placement by CoM state in the foot placement constrained condition as compared to steady-state walking (H2). When testing the $R^2$ for terminal swing, the best model included the factors *condition* and *speed* and their interaction *condition × speed*. There was extreme evidence for this model

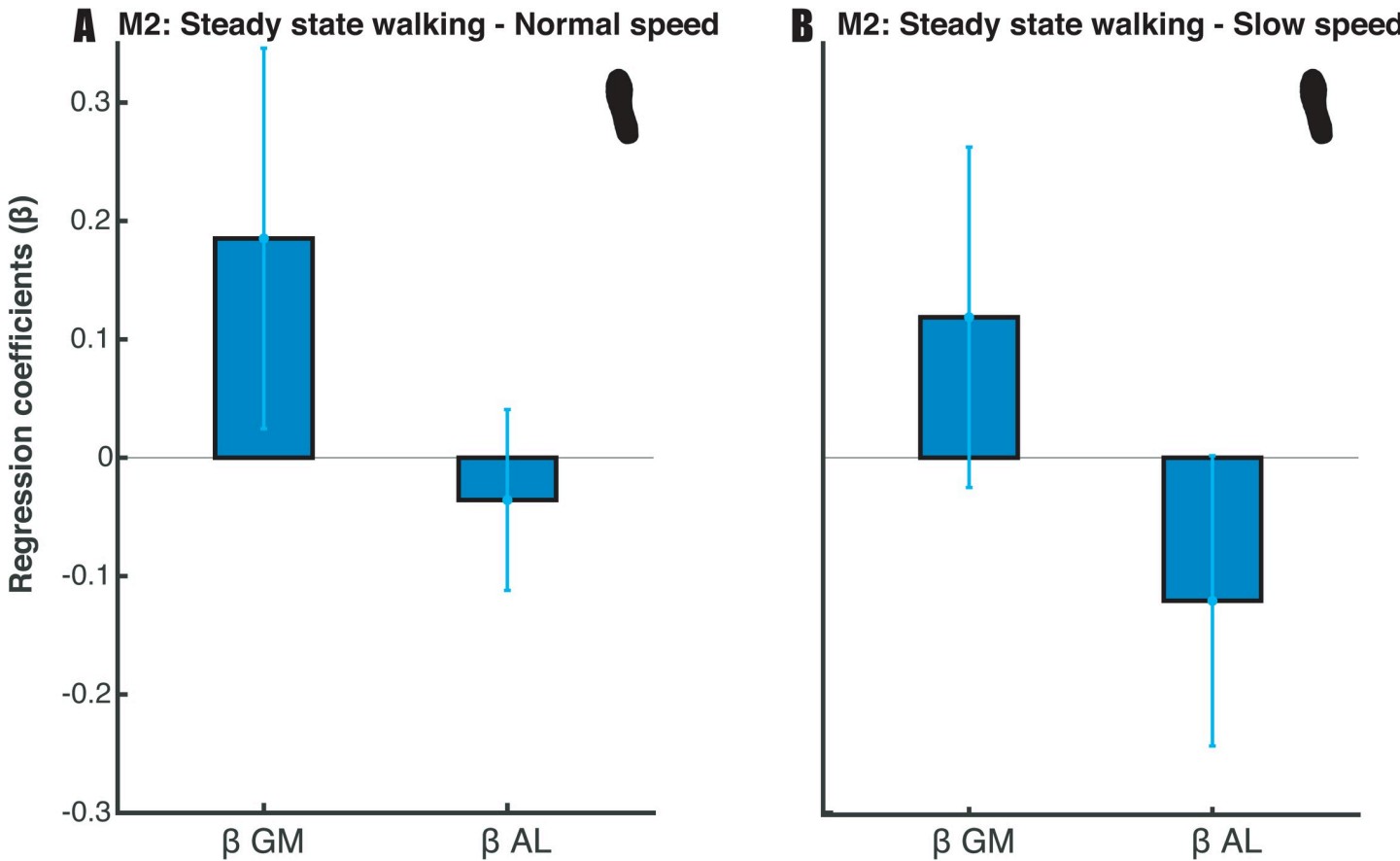

**Fig 6. Mean regression coefficients of the muscle model (2).** Standard deviations are represented by error bars. Panels A and B represent the results for respectively normal and slow walking speed. Moderate to extreme evidence ($BF_{10\_al}$ > 3 & $BF_{10\_gm}$ > 100) supports the inclusion of the predictors at normal walking speed. Extreme evidence ($BF_{10}$ > 100) supports the inclusion of the predictors at slow walking speed.

as compared to the Null model ($BF_{10\_ts}$ = 1.212·10$^{19}$). In view of the interaction effect, we conducted a one-tailed Bayesian paired samples t-test, for each speed separately. At normal walking speed we found moderate evidence ($BF_{10\_ts}$ = 7.352) in favor of a poorer prediction of foot placement by CoM state in the foot placement constrained condition as compared to steady-state walking (H2, Fig 9). At slow walking speed we found extreme ($BF_{10\_ts}$ = 2957.043) evidence supporting the $R^2$ to be lower in the foot placement constrained condition as compared to the steady-state walking condition.

## Discussion

We investigated the degree of (active) foot placement control during steady-state treadmill walking. We successfully replicated the findings of Wang and Srinivasan [4] and Rankin et al. [12] and can support that during steady-state walking foot placement is coordinated to CoM state and is associated with hip ab-/adductor muscle activity. The degree of foot placement control did not tighten when constraining the ankle moments. However, we found that the control strategy can be relaxed, achieving less tight foot placement control when constraining foot placement at a slow walking speed, whereas at a normal walking speed the degree of foot placement control was upheld. Overall, we can underscore the growing body of literature [3, 7,

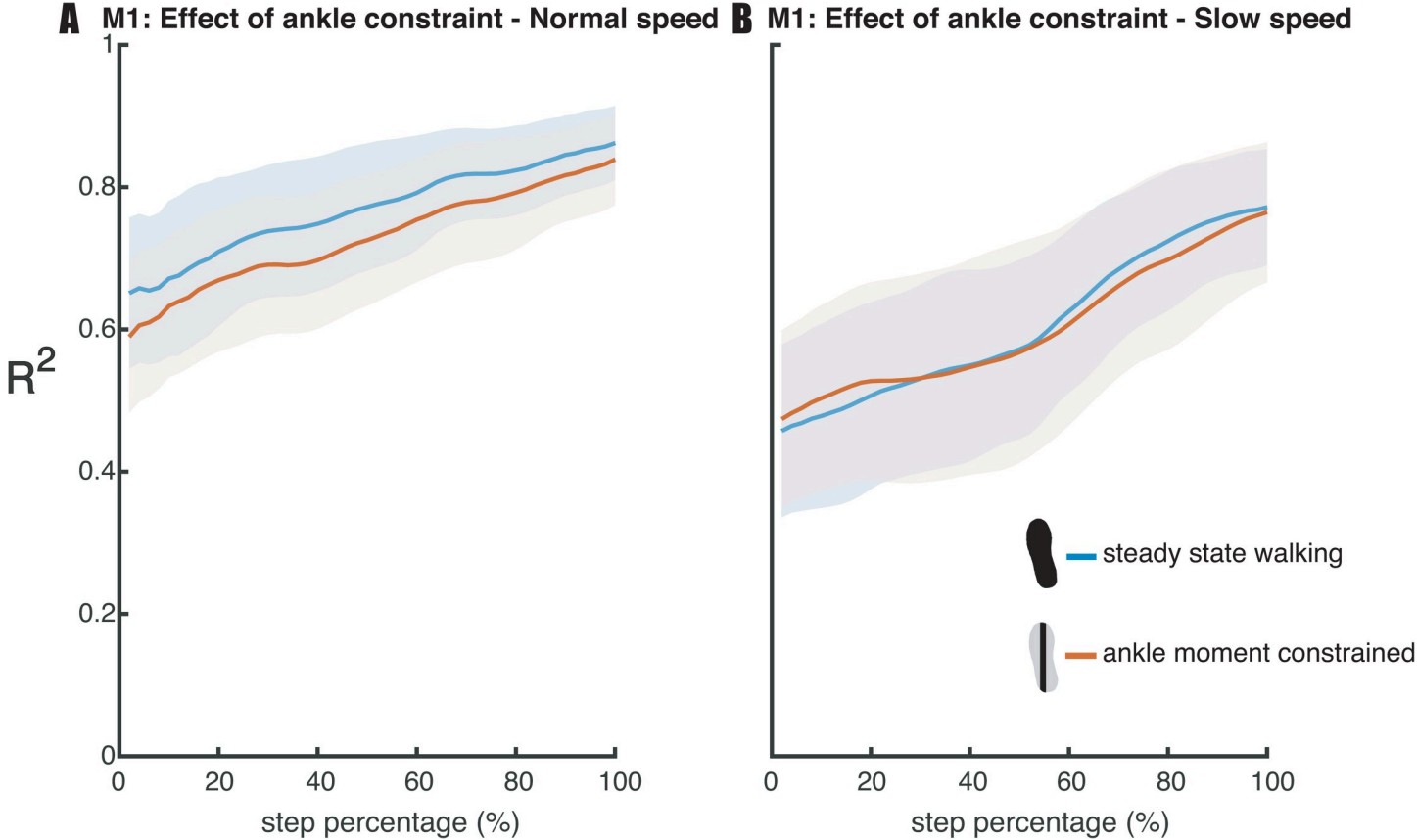

**Fig 7. Relative explained variance ($R^2$) of the foot placement model (1) during walking in the steady-state walking and ankle moment constrained condition.** Shaded areas depict the standard deviation. Panels A and B represent the results for respectively normal and slow walking speed. A step was defined from toe-off until subsequent heel strike.

8, 12, 15, 16, 26] and confirm that the foot placement model as introduced by Wang and Srinivasan [4] reflects an active control strategy, related to mediolateral stability [5].

### Ankle moment constrained walking

The ankle moments were constrained by narrowing the surface area underneath the shoe. As a result, the CoP shift was limited to the width of a narrow beam (see S1 Fig for illustration). We hypothesized that foot placement would compensate for the imposed constraint, by tightening control. However, for both normal and slow walking, no difference was found in the relative explained variance of the foot placement model as compared to steady-state walking. The degree of foot placement control was not tightened for compensation. Admittedly, we focused on steady-state gait control, while related studies mostly studied reactive gait control. Yet, this finding was unexpected given previous ones in the literature [7, 8, 11, 27]. For example, Hof et al. [11] demonstrated more lateral steps in prosthetic legs to accommodate for a limited CoP shift. Similarly, Vlutters et al. [27], showed foot placement adjustments in response to antero-posterior perturbations when wearing CoP-shift-limiting 'pin shoes'. Reimann et al. [7] reported that in response to vestibular stimulation execution of the ankle and foot placement strategy in temporal succession acted as the balance response. This suggests that without complementary ankle moments, a wider step might have been required to accommodate illusory falls, in line with their modelling results. And, Fettrow et al. [8] showed an inverse relationship

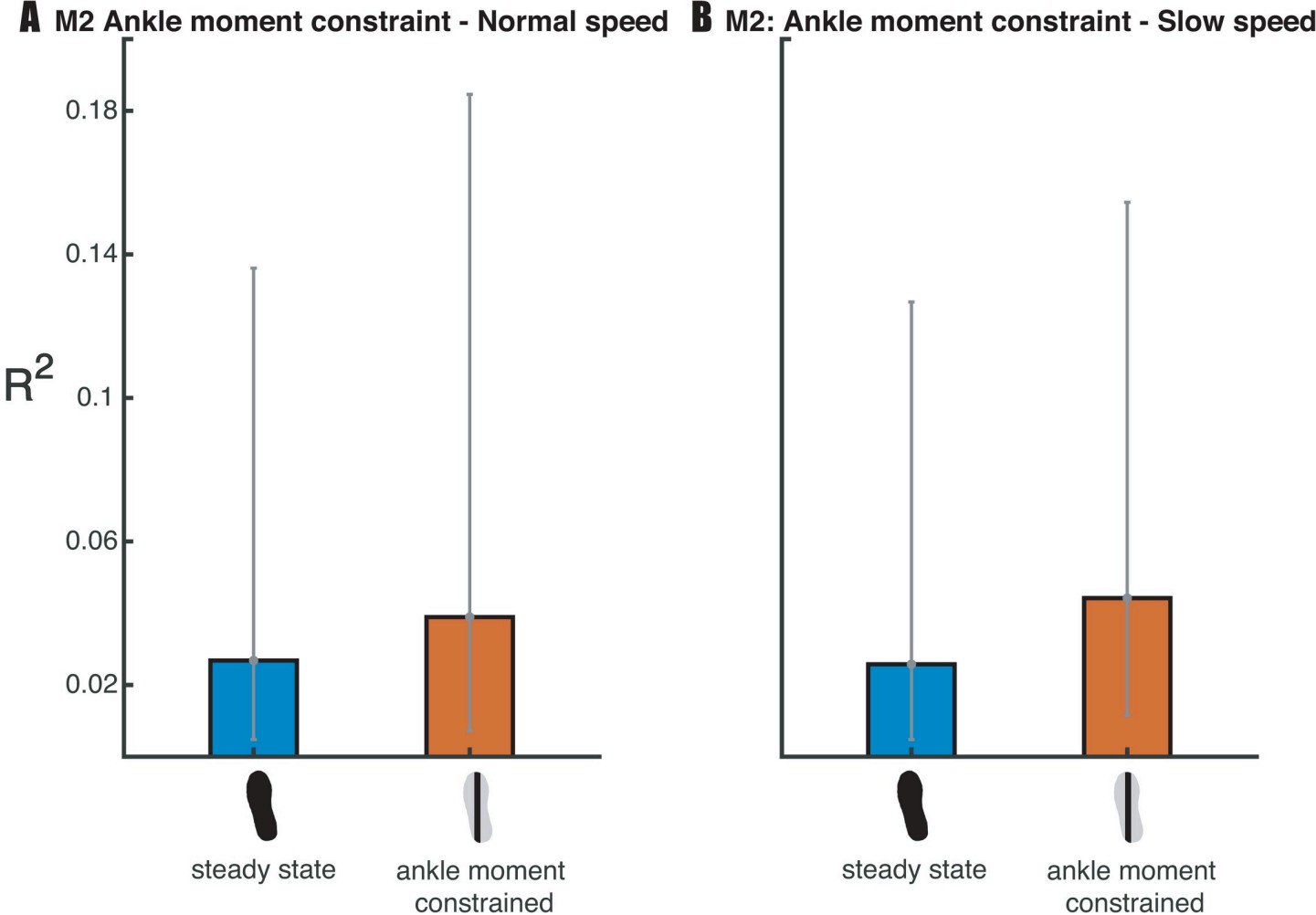

**Fig 8. Mean relative explained variance ($R^2$) of the muscle model (2) in the steady-state walking and the ankle moment constrained condition.** Standard deviations are represented by error bars. It remains inconclusive whether there is compensatory muscle activity in the ankle moment constrained condition.

between the execution of the ankle strategy and foot placement strategy during steady-state walking. Together these studies suggest that in response to natural variations or perturbations, foot placement control can accommodate for limited ankle moments. Here we would like to add that this does not necessarily imply more accurate control. An increased average step width can be used to maintain stability, with less tight foot placement control [28].

An important difference between the aforementioned studies and the current one is that their main outcome measure was based on average adjustments, i.e. on increases in step width or gain, whereas our $R^2$ outcome measure reflects foot placement accuracy in accommodating variations in CoM state. A limitation of our approach is that information provided by the intercept of the model is lost, i.e. the average step width. We hence explored differences in step width between the ankle moment constrained conditions and the steady-state walking conditions. Indeed, extreme evidence (BF > 100) demonstrated increased step width when participants walked with LesSchuh at both speeds (see S2 Fig). While this is likely to reflect a compensatory strategy, a limitation of our study could be that with LesSchuh participants should avoid to step into the middle of the split belt treadmill (as the surface area of the shoes was narrow enough to fit between the two belts). Participants might have increased their step

width as a precaution, and these findings should hence be interpreted with care. Nevertheless, increasing step width may be considered an appropriate response to the sustained (and invariable) perturbation of LesSchuh. Given the lower $R^2$ in the ankle moment constrained condition at normal walking speed (Fig 7), this perturbation may not only have perturbed the ankle moments, but foot placement as well. Probably, LesSchuh perturbed stance leg control during swing and consequently led to less accurate foot placement. This suggests that learning how to adapt to this perturbation may improve the degree of foot placement control, similar to earlier findings related to continuous foot placement perturbations [26].

An increased step width has earlier been considered as a general stabilizing strategy, characterizing cautious gait in an unpredictable situation [19, 28, 29]. An increase in step width is a possible temporary compensatory strategy that seems to be used until one is able to develop tighter control of foot placement. The duration of our trials (5 minutes at normal walking speed and 10 minutes at slow walking speed) might have been too short to adapt the degree of control. Recently, it has been shown that repeated exposure to a perturbing force field yielded an adaptation in foot placement control [26]. During later exposures tighter control was manifested as compared to the first 5-minute-perturbed trial. The initial increase in step widths was diminished in these later exposures. We conjecture that longer or multiple trials with LesSchuh can lead to an increased $R^2$ of the foot placement model. An increased degree of control may allow for a reduction of the average step width while maintaining stability. Walking with wider steps has been associated with a higher energy cost, and in normal walking individuals tend to select the step width that minimizes metabolic costs [30]. That is, adaptation over time and reduced step width may lead to a more economic compensatory strategy. Alternatively, increasing step width compared to normal walking might be a more economic strategy when walking with limited ankle moments. An additional energy cost related to actively increasing step-by-step control, might prevent participants to select narrower steps [31].

Further exploratory analysis in slow walking revealed that the imposed ankle moment constraint coincided with increased stride frequency, besides average step width, in spite of the metronome-imposed frequency (see S3 Fig). Increasing stride frequency has already been identified as a strategy to improve gait stability [32–34]. The use of an ankle strategy appears less prominent when walking with high as compared to low stride frequency [35]. By modulating stride frequency and increasing average step width, the need for more accurate foot placement control (as reflected by our outcome measure $R^2$) might have been circumvented. In future studies, it is worth looking into the hip and push-off strategies [2, 6, 8, 36, 37] to see whether they worked as compensatory strategies as well.

Whether compensatory hip ab-/adductor muscle activity does play a role in the compensatory response or not still remains inconclusive. We did not find a compensatory increase in step-by-step foot placement accuracy. Thus, it seems likely that hip ab-/adductor activity does not have a higher contribution to the variation in foot placement. However, the average increase in step width could have been driven by increased gluteus medius activity. This may explain why our analysis provided anecdotal (i.e. inconclusive) evidence rather than support for the null hypothesis. The low relative explained variance of the muscle model (2), although partly due to the noisy nature of EMG data, also suggests muscle activity alone cannot accurately predict foot placement. Likely passive dynamics during the swing phase have a large contribution, possibly controlled through push-off mechanisms [20, 36].

### Foot placement constrained walking

The foot placement constraint (lines projected on the treadmill) reduced step width variability in both slow and normal walking (see S4 Fig), denoting that the constraint was effective. An

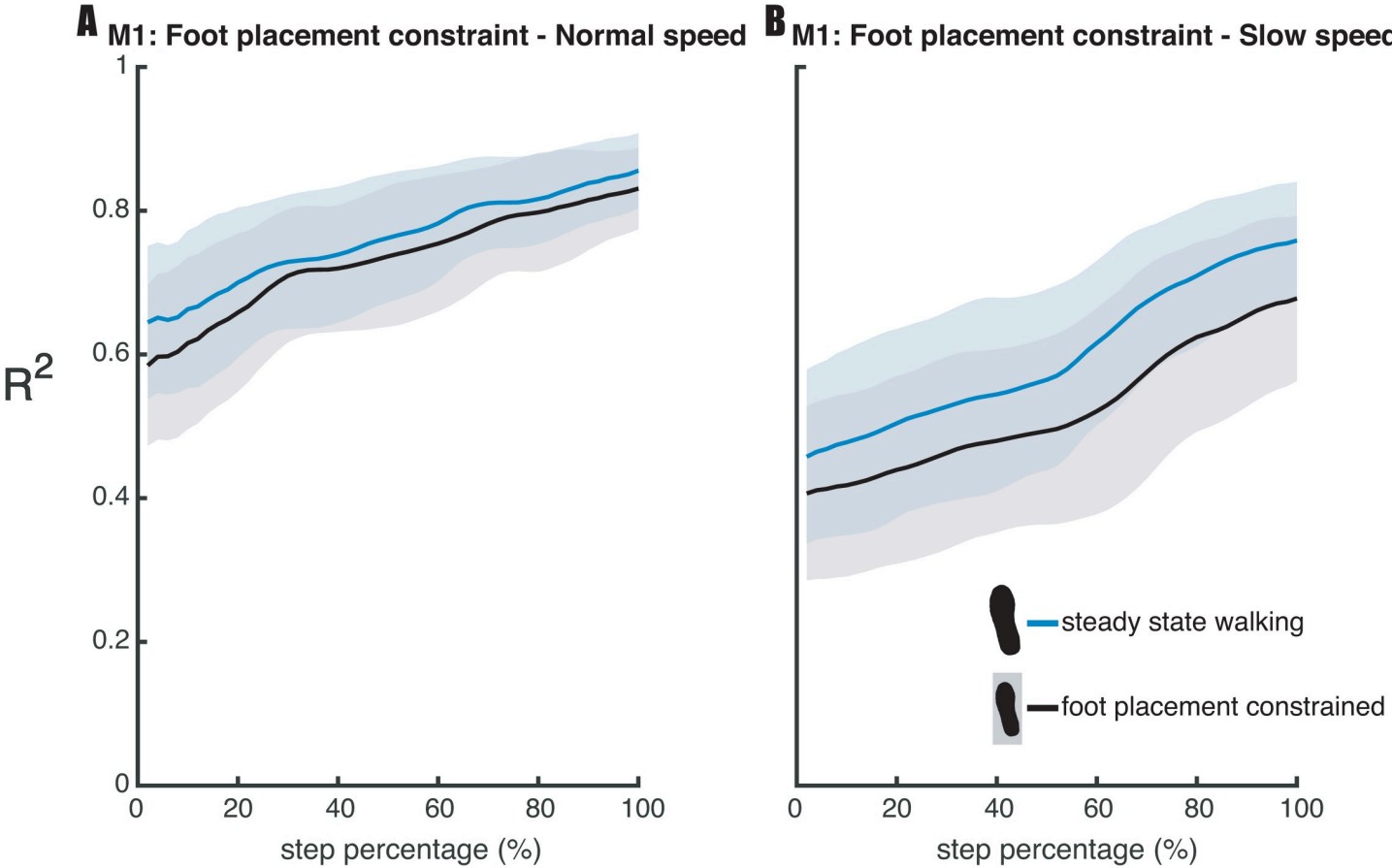

**Fig 9. Relative explained variance ($R^2$) of the foot placement model (1) in the steady-state walking and foot placement constrained condition.** Shaded areas depict the standard deviation. Panels A and B represent the results for respectively normal and slow walking speed. A step was defined from toe-off until subsequent heel strike.

earlier study already demonstrated reduced step width modulation by instructing participants to step on continuously projected lines on the treadmill [28]. In the current study the projected line appeared following toe-off. Therefore, we prevented preplanning a CoM trajectory during the preceding step [20, 38] in relation to the projection. Consequently, foot placement adjustments had to be realized during swing, which we expected to undermine the relationship between foot placement and CoM state. Evidence exists that rapid foot placement adjustments are feasible during a stepping task [39], although stability constraints appear to negatively affect responses to large jumps of foot placement targets [40]. In our study, foot placement appeared to be more effectively constrained by the projections during slow walking as compared to normal walking, as reflected by a lower step width variability at the slower speed (see S4 Fig). Regarding the degree of foot placement control, we found a *condition × speed* interaction effect on the $R^2$ of the foot placement model (model 1). The reduction in step width variability caused a diminished degree of foot placement control according to the foot placement model at a slow walking speed. At a normal walking speed this effect was less pronounced, and $R^2$ remained higher than during slow walking (Fig 9). In other words, at normal walking speed, the relationship between foot placement and CoM state was largely retained. This suggests that at a normal walking speed, stability constraints outweighed the task instruction. We interpret this to indicate that tight foot placement control is more important at normal as compared to slow walking speed. We would like to note that this agrees with previous findings that

the $R^2$ of the foot placement model is lower at slower speeds [13]. Yet, our results showed that foot placement remains actively controlled during slow steady-state walking, despite the lower degree of foot placement control. In contrast to an earlier study [28], we did not impose a step width different from average step width. However, from the earlier study we learn that the importance of tight foot placement control does not only differ with speed, but also given the average step width. At wider average step widths, participants demonstrated a lesser degree of step-by-step foot placement control [28]. Combined with the speed interaction effect in our study, these results demonstrate that the degree of foot placement control can be tightened/relaxed to satisfy stability demands under different task constraints.

## Conclusion

We found muscle driven step-by-step foot placement control during steady-state walking. This control appears to be more important at normal as compared to slow walking speed, based on the degree of foot placement control and adaptability to a foot placement constraint. When compensating for constrained ankle moments, average step width and stride frequency, rather than the degree of foot placement control were adjusted. Further research is required to unravel what other strategies might have contributed as a compensatory mechanism and whether the compensatory strategy is adapted over time. Perhaps longer exposure to walking with constrained ankle moments will lead to adoption of tighter foot placement control.

## Supporting information

**S1 Fig. Center of pressure shifts when walking with an unconstrained (normal shoe) left foot and an ankle moment constrained (LesSchuh) right foot.** The most medial (blue) and most lateral (red) shifts are plotted, showing divergence of these shifts when unconstrained (left panel) as compared to overlaying shifts when constrained (right panel). The mediolateral CoP shift is limited by a $\pm$ 1-centimeter ridge underneath LesSchuh (Fig 2). The figure presents an example of participant 18. Ankle moment constraint–effect of "LesSchuh".
(TIF)

**S2 Fig. Mean step width during the steady-state walking and ankle moment constrained conditions.** Blue and red bars represent respectively the steady-state walking and ankle moment constrained conditions. The grey lines connect the individual data points. An exploratory Bayesian repeated measures ANOVA, including the steady-state walking and ankle moment constrained condition at both speeds, revealed that the best model included only the factor "Condition" with extreme evidence as compared to the Null model ($BF_{10} = 1.610 * 10^{15}$). Post-hoc analysis provided extreme evidence supporting an increase in step width at both speeds to compensate for the ankle moment constraint ($BF_{10} = 1.064*10^{13}$). The influence of the ankle moment constrained condition on step width.
(TIF)

**S3 Fig. Mean stride frequency during the steady-state walking and ankle moment constrained conditions.** Blue and red bars represent respectively the steady-state walking and ankle moment constrained conditions. The grey lines connect the individual data points. As an exploratory analysis, as well as a protocol check, Bayesian repeated measures ANOVA, including the steady-state walking and ankle moment constrained condition at both speeds, revealed that the best model included the factors "Condition" and "Speed". Post-hoc analysis provided extreme evidence ($BF_{10} = 7.959*10^{42}$) indicating that stride frequency increased in the ankle moment constrained conditions as compared to steady-state walking. The influence

of the ankle moment constrained condition on stride frequency.
(TIF)

**S4 Fig. Mean step width variability during the steady-state walking and foot placement constrained conditions.** Blue and black bars represent respectively the steady-state walking and foot placement constrained conditions. The grey lines connect the individual data points. Bayesian repeated measures ANOVA, including the steady-state walking and foot placement strategy constrained condition at both speeds, revealed that the best model included the factors "Condition" and "Speed", with extreme evidence as compared to the Null model ($BF_{10}$ = 590646.967). Post-hoc analysis provided extreme evidence for a lower step width variability in the foot placement constrained condition as compared to steady-state walking ($BF_{10}$ = 55714.494). When comparing between speeds, a two-tailed Bayesian paired samples t-test provided extreme evidence demonstrated that in the foot placement constrained condition, the step width variability remained higher at a normal walking speed as compared to the slow walking speed ($BF_{10}$ = 2091.388). Effectiveness of the foot placement constraint: step width variability.
(TIF)

## Acknowledgments

The authors are thankful for all technical support and assistance during the data collection, especially for Leon Schutte, who built LesSchuh.

## Author Contributions

**Conceptualization:** A. M. van Leeuwen, J. H. van Dieën, S. M. Bruijn.

**Formal analysis:** A. M. van Leeuwen, S. M. Bruijn.

**Funding acquisition:** S. M. Bruijn.

**Investigation:** A. M. van Leeuwen.

**Methodology:** A. M. van Leeuwen, J. H. van Dieën, S. M. Bruijn.

**Supervision:** J. H. van Dieën, A. Daffertshofer, S. M. Bruijn.

**Visualization:** A. M. van Leeuwen, S. M. Bruijn.

**Writing – original draft:** A. M. van Leeuwen.

**Writing – review & editing:** A. M. van Leeuwen, J. H. van Dieën, A. Daffertshofer, S. M. Bruijn.

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
