## [Decision Letter · Decision Letter 0]

24 Aug 2020

PONE-D-20-17797

Step width and frequency to modulate: Active foot placement control ensures stable gait

PLOS ONE

Dear Dr. Bruijn,

Thank you for submitting your manuscript to PLOS ONE. After careful consideration, we feel that it has merit but does not fully meet PLOS ONE’s publication criteria as it currently stands. Therefore, we invite you to submit a revised version of the manuscript that addresses the points raised during the review process.

It has been reviewed carefully and the reviewers have provided some suggestions for improvement. If you could please address these concerns, and provide the journal with a point-by-point rebuttal as described below, I would very much like to see it again. Please note that if you feel that any of the concerns are beyond the scope of the work, you may say this in the rebuttal.

We look forward to receiving your revised manuscript.

Kind regards,

J. Lucas McKay, Ph.D., M.S.C.R.

Academic Editor

PLOS ONE

Journal Requirements:

Reviewers' comments:

Reviewer's Responses to Questions

**Comments to the Author**

1. Is the manuscript technically sound, and do the data support the conclusions?

Reviewer #1: Yes

Reviewer #2: Yes

2. Has the statistical analysis been performed appropriately and rigorously? 

Reviewer #1: Yes

Reviewer #2: Yes

3. Have the authors made all data underlying the findings in their manuscript fully available?

Reviewer #1: Yes

Reviewer #2: Yes

4. Is the manuscript presented in an intelligible fashion and written in standard English?

Reviewer #1: No

Reviewer #2: Yes

5. Review Comments to the Author

Reviewer #1: This manuscript reports the results of one experiment designed to investigate the mechanisms for maintaining gait stability. The study builds upon previous research suggesting that step-by-step variations in the mediolateral position of the foot play a key role in gait stability but that other strategies (e.g., the ankle push-off strategy) play a complementary role. The present study examines the interplay between these strategies by exploring how the degree of foot placement control varies as a function of constraints placed on ankle moments and foot placement. The results support the hypothesis that the degree of foot placement control is dependent on foot placement constraints. However, constraints on ankle moment did not affect the degree of foot placement control.

Overall, this is a solid study that makes a nice empirical contribution that advances our understanding of the role of foot placement control in gait stability. On the positive side, the rationale for the study is solid, the methodological approach is sound, the data are properly analyzed, and the interpretation of findings is reasonable. The authors preregistered their study, identified deviations from the preregistered plan, and will make their data available. There are a few issues that I think should be addressed before a final decision about this manuscript is made.

(1) The logic of the experiment appears to be sound but was not clearly presented. I believe that I was able to figure it out, but it required more effort than should be necessary. The basic aim of the study and the hypotheses are clearly explained in the introduction but the specific predictions are missing. What would be useful is a set of statements that connect the hypotheses (H1, H2, H3) with specific predictions for the analyses that were conducted (e.g., “If H# is true, then the R^2 of Model # should increase/decrease.”).

(2) I understand that the primary manipulations were the ankle moment and foot placement constraints, but it would be helpful if the authors could clarify the purpose of the walking speed manipulation and the predictions associated with this manipulation.

(3) Hypotheses 1 and 3 seem to capture the same idea. What is the difference? Is it possible for one to be true and the other false? Deriving specific predictions (as described in the previous comment) from both hypotheses would help to address this issue. (I wrote this comment after reading the Introduction and Methods. The difference eventually becomes clear to the reader but not until the Results section. The authors should clarify the difference earlier in the manuscript.)

(4) It would help the reader if the authors could begin each section of the results by explaining the question that each analysis is intended to answer.

(5) The manuscript has some grammatical issues, some of which are listed below. These should be addressed in the revision.

SPECIFIC COMMENTS AND MINOR ISSUES

Title: The title is awkwardly worded. I’m not sure what they mean by “Step width and frequency to modulate”.

Line 91: “vis-à-vis” is not properly used in this sentence. I think what the authors mean is “rather than”.

Line 92: Likewise, “accumulates” is not properly used.

Line 141: Replace “conform” with “in accordance with”.

Line 149: Add “phase” after familiarization.

Line 151: “Participants *were* familiarized with…”

Line 153: “…five *minutes*…”

Line 165: Did subjects receive any feedback about the accuracy of their footsteps?

Line 285: Delete “finally”

Reviewer #2: Review of “Step width and step frequency to modulate: active foot placement control ensures stable gate”

My summary of the paper:

In this paper, the author inquires how active foot placement control is compensated when the ability to modulate ankle moment is constrained or when step width is constrained. The author finds that instead of any modulation in foot placement control, the average step width and step frequency are changed. As the author admits, this does not necessarily mean that active foot placement does not adapt when constrained, because the ankle moment constraint imposed here could have inadvertently constrained the ability for active foot placement as well. The paper also replicates two results from past papers.

A summary of my review:

The paper is well-written and the results are discussed and interpreted with care and insightfully, in the context of the literature. Also, it is to be appreciated that the author appropriately preregistered their methods and declared deviations from preregistrations. It is great that the author will make the data available. One main suggestion I have to the author is to explore if there were any systematic trends in the CoM and foot placement states, which needed to be accounted for when demeaning. Secondly, I suggest the author compare the step width constrained control results to Perry and Srinivasan 2017, which doe a similar thing. In the writing, some further details and justifications need to be provided in the methods section of the paper and I provide specific suggestions for these below.

Detailed review (key changes suggested):

When the author says that the hip states and foot placement variables were “demeaned”, do they account for any systematic trends in the states over time? If not, I suggest the author de-trend these variables and perform the regression models again after de-trending to see if this changes the results. This is essential because, if there is a systematic trend over time in these states, it will affect the central results (the control gains) in this paper. I believe this will not take much time to add as it will just need the addition of one line of code.

The constraint condition in which the step width is constrained is similar to the constraint in Perry and Srinivasan 2017. I think it would be appropriate for the author compare and discuss the results here in comparison the results in that paper.

There are some additional details and justifications needed in the methods section. (i) Why did the author choose to constrain the step frequency? Neither of the papers this paper replicates use a metronome (Wang et al. and Rankin et al.). So, the author should justify this experimental choice to constrain step frequency. Also, it is unclear if the metronome was used for all the conditions i.e. steady-state, ankle constraint, and foot placement constraint. (ii) The author use a certain method to calculate the CoM (using regressions), could they provide more details about how the CoM was calculated or refer to a previous paper that calculates the CoM using a similar method? (iii) Further details need to be provided on the setup used for projecting beams onto the treadmill for the foot placement constraint condition. (iv) I suggest the author add a glossary explaining all the variables used in the paper in a single location.

Line-by-line comments (mostly minor):

Title: Can the title be revised for clarity? “step width and step frequency to modulate” sounds a bit unclear. Also, the focus of the paper is on the added constraints so that should feature in the tile in some way.

Abstract:

In line 20, “ankle strategy” is unclear. Maybe say “ankle moment modulation strategy” to show the link to the previous sentence.

In line 31, I think it is misleading to say “foot placement control was not tightened”, because a change in average step width and step frequency (as was found here) is also achieved through foot placement control. Maybe revise to “step-by-step foot placement control was not tightened”.

Introduction:

Very well-written. No changes suggested.

Methods:

Line 141, what is “conform Hof”?

Line 217, I assume COMpos and COMvel are vectors consisting of x, y, and z components? Please elaborate on this.

Line 224, there is a superscript “|” over the word “seconds” and it is not clear what this means. This shows up in other places in the paper as well. Please clarify.

Line 231, contralateral foot is also at midstance?

Line 233, stance foot is also at midstance? In general, please make sure that if states are being measured at a particular gait event, that is mentioned clearly.

Line 233, not sure what the word “predictors” refers to.

Line 238, why is the EMG variable subscripted “swing”? Was the EMG only calculated at swing? The result figures with EMG indicate it was calculated throughout the stride cycle.

Please state If FP2 and the EMG variables in equation 2 are demeaned or not somewhere in the methods section.

In equation 1, if CoM_pos(i) and CoM_vel(i) are written with (i) to indicate step cycle i, should FP also not be FP(i) and not just FP? I have a similar question about equations 2: to my understanding, the EMG are also for one step cycle, so should that also not be (i) ?

Results:

Figure 3 caption states that medial burst is greater than lateral burst, is this difference significant? It looks very small from the figure.

In figures 3 and 4, the y axis does not have any number scale.

Figure 4 caption says “higher emg activity during early swing”, but this appears to be true only for slow speed. Please clarify this in the caption itself.

Line 335, please state the R^2 value there itself. Did Rankin et al. have a similarly low R^2? Please discuss what this could mean.

Discussion:

The discussion is generally very well written and the interpretations are made carefully.

In line 395, maybe use the word “relaxed” instead of “adapted”, to contrast with “tightened"?

Line 414, unclear what “serially coordinated as a balance response” means, please clarify

Line 420, could you elaborate on why you say “this does not necessarily imply more accurate control”

Conclusion:

In line 511, the use of the word “other” seems to indicate that foot placement control compensation was clearly eliminated in this study. However, as the author admits, this is not the case. It is possible that the LeSchuh did not allow for easy foot placement modulation. This can be explored with further experiments to see if the foot placement control compensates with practice with LeSchuh. Please add this caveat in the conclusion.

6. PLOS authors have the option to publish the peer review history of their article (what does this mean?). If published, this will include your full peer review and any attached files.

Reviewer #1: No

Reviewer #2: No

---

## [Author Response · Author response to Decision Letter 0]

19 Oct 2020

Reviewer #1: This manuscript reports the results of one experiment designed to investigate the mechanisms for maintaining gait stability. The study builds upon previous research suggesting that step-by-step variations in the mediolateral position of the foot play a key role in gait stability but that other strategies (e.g., the ankle push-off strategy) play a complementary role. The present study examines the interplay between these strategies by exploring how the degree of foot placement control varies as a function of constraints placed on ankle moments and foot placement. The results support the hypothesis that the degree of foot placement control is dependent on foot placement constraints. However, constraints on ankle moment did not affect the degree of foot placement control.

Overall, this is a solid study that makes a nice empirical contribution that advances our understanding of the role of foot placement control in gait stability. On the positive side, the rationale for the study is solid, the methodological approach is sound, the data are properly analyzed, and the interpretation of findings is reasonable. The authors preregistered their study, identified deviations from the preregistered plan, and will make their data available. There are a few issues that I think should be addressed before a final decision about this manuscript is made.

We wish to thank the reviewer for their kind words and helping us to solve remaining issues. We have adapted the manuscript based on the comments raised. Below, we provide a point-by-point reply to each reviewer comment. In addition, we highlighted all changes made in the manuscript. 

(1) The logic of the experiment appears to be sound but was not clearly presented. I believe that I was able to figure it out, but it required more effort than should be necessary. The basic aim of the study and the hypotheses are clearly explained in the introduction but the specific predictions are missing. What would be useful is a set of statements that connect the hypotheses (H1, H2, H3) with specific predictions for the analyses that were conducted (e.g., “If H# is true, then the R^2 of Model # should increase/decrease.”).

We understand the need for clarification and added:

 ‘…(i.e. a higher R2 of model (1))…’ (Page 18, Lines 360-361)

‘…(i.e. a higher R2 of model (2))…’ (Page 18, Lines 361-362)

‘…(i.e. a lower R2 of model (1))…’ (Page 18, Line 371)

to specify our predictions.

(2) I understand that the primary manipulations were the ankle moment and foot placement constraints, but it would be helpful if the authors could clarify the purpose of the walking speed manipulation and the predictions associated with this manipulation.

At a slower walking speed, the degree of foot placement control has been shown to be lower [1]. Apparently, the importance of foot placement control differs at different walking speeds. This could have affected the influence of our other manipulations. We therefore felt that testing at two walking speeds would strengthen our conclusions. We did not however make any specific speed-related predictions regarding foot placement control.

We have now tried to more explicitly state these ideas in the manuscript:

“We did not make any speed-related predictions, but the speed-dependent nature of the foot placement strategy could potentially affect the effects of our other experimental conditions.” (Page 7, Lines 127-129)

Apart from a focus on foot placement control, we also made predictions regarding the mediolateral ankle strategy under the same conditions. This is still work in progress, so for now we refer to the preregistration (https://osf.io/74pn5) for speed-related predictions which were not incorporated in this paper. 

(3) Hypotheses 1 and 3 seem to capture the same idea. What is the difference? Is it possible for one to be true and the other false? Deriving specific predictions (as described in the previous comment) from both hypotheses would help to address this issue. (I wrote this comment after reading the Introduction and Methods. The difference eventually becomes clear to the reader but not until the Results section. The authors should clarify the difference earlier in the manuscript.)

We now specify our predictions clearer in the Methods:

“That is, this allowed for testing the hypotheses that compensation will encompass tighter control (i.e. a higher R2 of model (1)) (H1) driven by compensatory muscle activation (i.e. a higher R2 of model (2)) (H3).” (Page 18, Lines 359-362)

Furthermore, we now explain the relation between these two hypotheses as follows:

 “If both H1 and H3 are true, this would suggest a compensatory tighter coupling between variations in foot placement and CoM state (H1), achieved through compensatory muscle activity (H3). If only H1 is true, this would suggest tighter coupling between variations in foot placement and CoM state (H1), but no evidence that this is actively controlled (in contrast with H3). If only H3 is true, this would suggest that foot placement is more strictly controlled through muscle activity (H3), but not to tighten the coupling between variations in foot placement and CoM state.” (Page 18, Lines 362-368)

(4) It would help the reader if the authors could begin each section of the results by explaining the question that each analysis is intended to answer.

We have added the following to address this comment:

 “First, we considered the relative explained variance (R2) and tested the regression coefficients of the foot placement model (1) to assess our expectation (E1) that mediolateral foot placement can be predicted by CoM state, as previously shown by Wang and Srinivasan [4].” (Page 20, Lines 398-400)

 “Our second expectation (E2), that mediolateral foot placement correlates with hip ab- and adductor muscle activity during the preceding swing phase, was tested based on the regression coefficients of the muscle model (2), and visualized in Fig 3 and Fig 4 below.” (Page 21, Lines 409-411)

 “Regarding our first hypothesis (H1), that constrained ankle moments would lead to tighter foot placement control, we tested the relative explained variance (R2) of the foot placement model (1).” (Page 24, Lines 437-439)

 “Regarding our third hypothesis (H3), that constrained ankle moments would lead to larger active contribution to step-by-step variability in mediolateral foot placement, we tested the relative explained variance (R2) of the muscle model (2).” (Page 25, Lines 450-452)

“Regarding our second hypothesis (H2), that the degree of foot placement control would decrease, when constraining foot placement, we tested the relative explained variance (R2) of the foot placement model (1).“ (Page 26, Lines 465-467)

The manuscript has some grammatical issues, some of which are listed below. These should be addressed in the revision.

SPECIFIC COMMENTS AND MINOR ISSUES

(1) Title: The title is awkwardly worded. I’m not sure what they mean by “Step width and frequency to modulate”.

Also considering the other reviewer’s comments we now rephrased our title to be:

Active foot placement control ensures stable gait: Effect of constraints on foot placement and ankle moments

Line 91: “vis-à-vis” is not properly used in this sentence. I think what the authors mean is “rather than”.

We have replaced “vis-à-vis” with “rather than” (Page 8, Line 156)

(2) Line 92: Likewise, “accumulates” is not properly used.

“though there is increasing” (Page 8, Line 157)

(4) Line 141: Replace “conform” with “in accordance with”.

We have replaced this in the manuscript. (Page 11, Lines 205-206)

(5) Line 149: Add “phase” after familiarization.

We have added this in the manuscript. (Page 11, Line 216)

(6) Line 151: “Participants *were* familiarized with…”

We have added this in the manuscript. (Page 11, Line 218)

(7) Line 153: “…five *minutes*…”

We have added this in the manuscript, (Page 12, Line 220)

(8) Line 165: Did subjects receive any feedback about the accuracy of their footsteps?

They did not receive any feedback other than that they could see themselves where they stepped.

(9) Line 285: Delete “finally”

We have deleted this from the manuscript.

Reviewer #2: Review of “Step width and step frequency to modulate: active foot placement control ensures stable gate”

My summary of the paper:

In this paper, the author inquires how active foot placement control is compensated when the ability to modulate ankle moment is constrained or when step width is constrained. The author finds that instead of any modulation in foot placement control, the average step width and step frequency are changed. As the author admits, this does not necessarily mean that active foot placement does not adapt when constrained, because the ankle moment constraint imposed here could have inadvertently constrained the ability for active foot placement as well. The paper also replicates two results from past papers.

A summary of my review:

The paper is well-written and the results are discussed and interpreted with care and insightfully, in the context of the literature. Also, it is to be appreciated that the author appropriately preregistered their methods and declared deviations from preregistrations. It is great that the author will make the data available. One main suggestion I have to the author is to explore if there were any systematic trends in the CoM and foot placement states, which needed to be accounted for when demeaning. Secondly, I suggest the author compare the step width constrained control results to Perry and Srinivasan 2017, which does a similar thing. In the writing, some further details and justifications need to be provided in the methods section of the paper and I provide specific suggestions for these below.

We wish to thank the reviewer for their kind words and thoughtful suggestions. We have addressed these suggestions in a point-by-point reply below. In addition, we highlighted all changes made in the manuscript. 

Detailed review (key changes suggested):

(1) When the author says that the hip states and foot placement variables were “demeaned”, do they account for any systematic trends in the states over time? If not, I suggest the author de-trend these variables and perform the regression models again after de-trending to see if this changes the results. This is essential because, if there is a systematic trend over time in these states, it will affect the central results (the control gains) in this paper. I believe this will not take much time to add as it will just need the addition of one line of code.

We understand detrending could be essential given a drifting position of the participant on the treadmill, and value your concern. However, since we express both the center of mass as well as foot placement with respect to the contralateral stance foot prior to demeaning, we think it is not essential in this case. 

To clarify that the demeaning was done after expressing CoMpos and FP with respect to the contralateral stance foot at midstance, we now first describe the variables (i.e. their expression with respect to the contralateral stance foot) in lines 306-312 (pages 15-16), while leaving “demeaning” out of these sentences. Subsequently, at the end of that paragraph we now state:

“FP, CoMpos and CoMvel were demeaned prior to regression.” (Page 16, Line 310)

Moreover, in response to another reviewer comment, we now added a glossary (Pages 2-4, Lines 14-73) to present our variables. In the glossary we mention for the CoM position (CoMpos):

“Mediolateral CoM position (along the global x-axis) expressed with respect to mediolateral position of the stance foot at midstance.” (Page 2, Lines 29-30)

and for foot placement (FP):

“The mediolateral (along the global x-axis) foot position (calcaneus position digitized with respect to the foot cluster marker) at midstance expressed with respect to the mediolateral position of the contralateral foot at midstance (i.e. step width).” (Page 2, Lines 23-286)

We hope these additions clarify that we expressed the variables with respect to the contralateral stance foot, rather than in the global coordinate system. Based on this we believe detrending is not necessary.

(2) The constraint condition in which the step width is constrained is similar to the constraint in Perry and Srinivasan 2017. I think it would be appropriate for the author compare and discuss the results here in comparison the results in that paper.

Thank you for making us aware of this paper. We have added this to our discussion as follows:

 “An earlier study already demonstrated reduced step width modulation by instructing participants to step on continuously projected lines on the treadmill [28].” (Page 32, Lines 584-586)

as well as

“In contrast to an earlier study [28], we did not impose a step width different from average step width. However, from the earlier study we learn that the importance of tight foot placement control does not only differ with speed, but also given the average step width. At wider average step widths, participants demonstrated a lesser degree of step-by-step foot placement control [28]. Combined with the speed interaction effect in our study, these results demonstrate the degree of foot placement control can be tightened/relaxed to satisfy stability demands given different task constraints.” (Page 33, Lines 606-613)

Furthermore, we have added this paper as a reference for the following statement we made before:

“An increased step width has earlier been considered as a general stabilizing strategy, characterizing cautious gait in an unpredictable situation [19, 28, 29].” (Page 30, Lines 542-543)

Moreover, we cited this paper to underscore that not only step-by-step foot placement control, but also average step width promotes (mediolateral) gait stability. We did so in the following statement.

 “An increased average step width can be used to maintain stability, with less tight foot placement control [28].” (Page 29, Lines 519-520)

(3) There are some additional details and justifications needed in the methods section. 

(i) Why did the author choose to constrain the step frequency? Neither of the papers this paper replicates use a metronome (Wang et al. and Rankin et al.). So, the author should justify this experimental choice to constrain step frequency. Also, it is unclear if the metronome was used for all the conditions i.e. steady-state, ankle constraint, and foot placement constraint. 

We wanted to constrain stride frequency as we knew from earlier studies [2-4] that varying step frequency is a stability control strategy. More specific, gait can be made more robust by increasing stride frequency. Since we wanted to evaluate the effect of the different constraints on the degree of foot placement control, we attempted to keep potential confounding factors, such as stride frequency constant. We have clarified this in the manuscript as follows:

“In this study, stride frequency control could be a confounding stability control strategy [19]. Therefore, stride frequency was controlled by means of a metronome, to avoid that participants changed stride frequency between conditions.” (Page 11, Lines 206-208)

(ii) The author use a certain method to calculate the CoM (using regressions), could they provide more details about how the CoM was calculated or refer to a previous paper that calculates the CoM using a similar method? 

We are happy that you pointed out this mistake in writing. We estimated the segment’s mass using a regression. The CoM however was expressed as a percentage of the longitudinal axis of the segment. We have adjusted the manuscript to include this as follows:

 “we estimated the segment’s mass via linear regression including the segment’s length and the segment’s measured circumference as predictors and regression coefficients based on gender [24]. The segment’s CoM was estimated as a percentage of the longitudinal axis of the segment [24, 25].” (Pages 14-15, Lines 284-287)

(iii) Further details need to be provided on the setup used for projecting beams onto the treadmill for the foot placement constraint condition.

We have added more detail to the paragraph below, as highlighted in the manuscript:

“Projections on the treadmill served to constrain the variation in mediolateral foot placement. In brief, first the average step width was derived from the final 100 steps of the familiarization trial, based on the CoP estimated from the force measurement of the instrumented treadmill. Then, this average step width was imposed by projecting beams on the treadmill, and participants were instructed to place their foot in the middle of the beam. For every step, the beam became visible following toe-off in order to prevent modification of the CoM swing phase trajectory by compensatory push-off modulation [20]. Customized labview software allowed us to estimate the toe-off event based on a force threshold, which triggered the projections based on real-time force measurements.” (Page 12, lines 227-235)

 (iv) I suggest the author add a glossary explaining all the variables used in the paper in a single location.

Thank you for this suggestion, we feel this overview clarifies our paper a lot. In the manuscript you can find the following glossary following the title page: 

“Glossary

CoM Center of mass.

CoP Center of pressure.

BF10 Bayes factor indicating evidence supporting the alternative hypothesis. 

BF01 Bayes factor indicating evidence supporting the null hypothesis.

FP The mediolateral (along the global x-axis) foot position (calcaneus position digitized with respect to the foot cluster marker) at midstance expressed with respect to the mediolateral position of the contralateral foot at midstance (i.e. step width). The variable was demeaned prior to the performed regression (model 1).

CoMpos Mediolateral CoM position (along the global x-axis) expressed with respect to mediolateral position of the stance foot at midstance. This variable was demeaned prior to the performed regression (model 1).

βpos Regression coefficient defining the relationship between mediolateral CoM position (CoMpos) and step width (FP), as part of the foot placement model (model 1).

CoMvel Mediolateral CoM velocity (along the global x-axis), calculated as the derivative of the CoM position expressed in the global coordinate system. This variable was demeaned prior to the performed regression (model 1). 

βvel Regression coefficient defining the relationship between mediolateral CoM velocity (CoMvel) and step width (FP), as part of the foot placement model (model 1).

FP2 The mediolateral (along the global x-axis) foot position (calcaneus position digitized with respect to the foot cluster marker) at midstance expressed with respect to the CoM at the time of toe-off in accordance with Rankin et al. [12]. The variable was demeaned and divided by the standard deviation prior to the performed regression (model 2).

EMGgm_swing The median gluteus medius EMG amplitudes over 60-80% of the gait cycle (early swing), multiplied by the duration of this episode in seconds. The variable was demeaned and divided by the standard deviation prior to the performed regression (model 2).

βgm_swing Regression coefficient defining the relationship between gluteus medius activity during early swing (EMGgm_swing) and step width (FP2), as part of the muscle model (model 2).

EMGal_swing The median adductor longus EMG amplitudes over 60-80% of the gait cycle (early swing), multiplied by the duration of this episode in seconds. The variable was demeaned and divided by the standard deviation prior to the performed regression (model 2).

βal_swing Regression coefficient defining the relationship between adductor longus activity during early swing (EMGgm_swing) and step width (FP2), as part of the muscle model (model 2).

R2 The relative explained variance of models (1) and (2). For model 1 this variable is interpreted as a measure of the degree of foot placement control. For model 2 this variable is interpreted as a measure of the active contribution to step-by-step foot placement control.“ (Pages 2-4, Lines 14-73)

Line-by-line comments (mostly minor):

(1) Title: Can the title be revised for clarity? “step width and step frequency to modulate” sounds a bit unclear. Also, the focus of the paper is on the added constraints so that should feature in the tile in some way.

We appreciate your comment and tried to clarify the title as follows:

Active foot placement control ensures stable gait: Effect of constraints on foot placement and ankle moments

Abstract:

(2) In line 20, “ankle strategy” is unclear. Maybe say “ankle moment modulation strategy” to show the link to the previous sentence.

Based on your suggestion we have replaced “ankle strategy” by “ankle moment control” in line 82.

(3) In line 31, I think it is misleading to say “foot placement control was not tightened”, because a change in average step width and step frequency (as was found here) is also achieved through foot placement control. Maybe revise to “step-by-step foot placement control was not tightened”.

We agree and have added the suggested ‘step-by-step’ (Page 5, Line 93)

Introduction:

Very well-written. No changes suggested.

Methods:

(4) Line 141, what is “conform Hof”?

Here we meant to refer to the paper by Hof (1996) in which recommendations are given to scale data to body size. Based on the suggestion of the other reviewer, we now rephrased this as 

“…walking speeds, normalized to leg length in accordance with Hof [18].” (Page 11, 205-206)

in the manuscript.

(5) Line 217, I assume COMpos and COMvel are vectors consisting of x, y, and z components? Please elaborate on this.

We were interested in the “mediolateral CoM position CoMpos” and the mediolateral CoM velocity CoMvel” (Page 15, Lines 290-291), and hence, the CoMpos and CoMvel are vectors containing only ML elements. 

We have now more clearly written this; 

 “Mediolateral CoM displacement was defined along the x-axis of our global coordinate system, of which the x-axis was oriented perpendicular to the direction of the treadmill.” (Page 15, Lines 288-290)

Lastly, we have emphasized this in the glossary as well (Pages 2-4, Lines 14,-73), specifically in the following lines:

“Mediolateral CoM position (along the global x-axis) …” (Page 2, Line 29)

And

“Mediolateral CoM velocity (along the global x-axis) ...” (Page 3, Lines 37)

(6) Line 224, there is a superscript “|” over the word “seconds” and it is not clear what this means. This shows up in other places in the paper as well. Please clarify.

The superscript refers to the deviations from the preregistered protocol, described later in the manuscript. We now clarified this by using brackets and writing

“(we note here that this is a deviation from our preregistered protocol, see “Deviations from the preregistered plans I”).” (Page 15, Lines 297-299)

and later 

“(we note here that this is a deviation from our preregistered protocol, see “Deviations from the preregistered plans II”).” (Page 16, Lines 317-318)

and

“(we note here that this is a deviation from our preregistererd protocol, see “Deviations from the preregistered plans III”).” (Page 17, Lines 342-343)

instead.

(7) Line 231, contralateral foot is also at midstance?

Indeed. We have clarified this in the manuscript by adding “…at midstance” (Page 15, Line 306).

(8) Line 233, stance foot is also at midstance? In general, please make sure that if states are being measured at a particular gait event, that is mentioned clearly.

Again we have added “…at midstance” (Page 15, Line 308) to clarify our procedures.

(9) Line 233, not sure what the word “predictors” refers to.

With “predictors” we mean the independent variables as introduced in the line before. To clarify this we have added “(i.e. predictors)” (Page 15, Line 308).

(10) Line 238, why is the EMG variable subscripted “swing”? Was the EMG only calculated at swing? The result figures with EMG indicate it was calculated throughout the stride cycle.

It is true that the figures show the EMG throughout the stride cycle. However, for the regression we included a measure for the EMG activity from 60-80% of the gait cycle, corresponding to early swing. Hence, we subscripted this variable “swing” for the regression. We have now mentioned this more clearly as “the median gluteus medius and adductor longus EMG amplitudes from 60-80% of the gait cycle” instead of just “gluteus medius and adductor longus EMG amplitudes” (Page 16, Line 314-315).

(11) Please state If FP2 and the EMG variables in equation 2 are demeaned or not somewhere in the methods section.

These variables were also demeaned, as now has been described more explicitly: “we normalized FP2, EMGgm_swing and EMGal_swing by demeaning and dividing by the standard deviation” (Page 16, Lines 316-317).

Moreover, these variables have been described in the glossary (Pages 2-4, Lines 14-73), as follows:

“EMGgm_swing The median gluteus medius EMG amplitudes over 60-80% of the gait cycle (early swing), multiplied by the duration of this episode in seconds. The variable was demeaned and divided by the standard deviation prior to the performed regression (model 2). (Page 3, Lines 52-55)

EMGal_swing The median adductor longus EMG amplitudes over 60-80% of the gait cycle (early swing), multiplied by the duration of this episode in seconds. The variable was demeaned and divided by the standard deviation prior to the performed regression (model 2).” (Page 4, Lines 61-64)

(12) In equation 1, if CoM_pos(i) and CoM_vel(i) are written with (i) to indicate step cycle i, should FP also not be FP(i) and not just FP? I have a similar question about equations 2: to my understanding, the EMG are also for one step cycle, so should that also not be (i) ?

Thank you for noting this mistake in the manuscript. In line 323 we stated “in every step cycle i”, which is wrong. “i” in this case is not a reference to the step cycle, but to an index representing the percentages of the step cycle. We normalized to 51 samples per step, allowing us to implement the model with predictors at different phases of the gait cycle. Therefore, we have rewritten this as:

“…for different phases (i) of the step cycle.” (Page 16, line 321)

Results:

(13) Figure 3 caption states that medial burst is greater than lateral burst, is this difference significant? It looks very small from the figure.

The figures are merely a dichotomous visual representation of the “significant” regression (note that we used Bayesian statistics and strictly cannot use the word significant in this case). In figures 3 and 4 we wanted to illustrate that the relationship between hip ab-/adductor activity and foot placement describes more lateral foot placement with a higher GM burst and a lower AL burst. We statistically tested the beta coefficients of the relationship against zero and found moderate to extreme evidence supporting both muscles predicted foot placement. Note that we did not directly test the differences as presented in the figures.

We agree the burst seem really small, although they are consistent with our expectations, the muscles’ function and earlier results reported by [5]. Despite this small difference we feel the figures illustrate nicely what the relationship represents and therefore we did not make any changes to the manuscript.

(14) In figures 3 and 4, the y axis does not have any number scale.

Since the unit of EMG amplitude is largely arbitrary, as it is affected by individual differences such as muscle mass and skin conductance, we normalized the EMG to the average stride peak activity (i.e. for the GM the large peak during stance) for each speed respectively. We now added a number scale. Since these normalized numbers do not contribute to our message or what we wanted to test, we added a clarification in the caption as follows:

 “The figure serves as a dichotomous illustration of the relationship established through regression and does not show values that were statistically tested.” (Pages 21-22, Fig 3&4, captions):

(15) Figure 4 caption says “higher emg activity during early swing”, but this appears to be true only for slow speed. Please clarify this in the caption itself.

By higher we meant the comparison between the medial and lateral steps, and only from 60-80 % of the gait cycle. Although this difference is clearer for the gluteus medius in figure 3, in figure 4 it holds for both speeds that the light blue line is mostly above the dark blue line during early swing. Indeed, based on the figure only, this seems clearer at a slow speed and a bit questionable at a normal speed. We again want to emphasize that the figures are merely a visualization, whereas our conclusions are based on the results of the regression.

However, to make the caption clearer we have added:

“When comparing medial to lateral steps during early swing (60-80% of the gait cycle),” (Page 22, Fig 4, caption) to draw the attention to the period and comparison of interest. 

“This is more prominent at slow walking speed.” (Page 22, Fig 4, caption) 

to acknowledge it is not that clear at a normal walking speed.

(16) Line 335, please state the R^2 value there itself. Did Rankin et al. have a similarly low R^2? Please discuss what this could mean.

Rankin et al. (2014) did not report the R^2 values in their original paper [5]. However, we corresponded with Dr, Dean (who is the corresponding author of Rankin et al., 2014 [5]). Although he asked me not to cite him on this (he preferred to do a more thorough check of the data to check the exact values they found). He told me their typical R^2 values were ~0.10. This value is within our range of R^2 values, although our mean R^2 is lower.

We have added “(Mnormal_speed = 0.0268; Mslow_speed =0.0258)” (Page 22, lines 429-430) to report the R^2 values.

Furthermore, in the Discussion we have added:

 “The low relative explained variance of the muscle model (2), although partly due to the noisy nature of EMG data, also suggests muscle activity alone cannot accurately predict foot placement. Likely passive dynamics during the swing phase have a large contribution, possibly controlled through push-off mechanisms [20, 36].” (Page 31, Lines 577-580)

Discussion:

The discussion is generally very well written and the interpretations are made carefully.

(17) In line 395, maybe use the word “relaxed” instead of “adapted”, to contrast with “tightened"?

Thank you for your suggestion we have replaced “adapted” by “relaxed” (Page 28, Line 493).

(18) Line 414, unclear what “serially coordinated as a balance response” means, please clarify

We have reformulated this in the manuscript as follows:

“…in temporal succession acted as a balance response.” (Page 29, Lines 512-513)

(19) Line 420, could you elaborate on why you say “this does not necessarily imply more accurate control”

We have clarified this statement by adding:

 “An increased average step width can be used to maintain stability, with less tight foot placement control [28].” (Page 29, Lines 519-520)

Conclusion:

(20) In line 511, the use of the word “other” seems to indicate that foot placement control compensation was clearly eliminated in this study. However, as the author admits, this is not the case. It is possible that the LeSchuh did not allow for easy foot placement modulation. This can be explored with further experiments to see if the foot placement control compensates with practice with LeSchuh. Please add this caveat in the conclusion.

We have added the following to the conclusion:

 “Perhaps longer exposure to walking with constrained ankle moments will lead to adoption of tighter foot placement control”. (Page 33, Lines 622-624)

References

1. Stimpson KH, Heitkamp LN, Horne JS, Dean JC. Effects of walking speed on the step-by-step control of step width. Journal of biomechanics. 2018;68:78-83.

2. Hak L, Houdijk H, Steenbrink F, Mert A, van der Wurff P, Beek PJ, et al. Stepping strategies for regulating gait adaptability and stability. Journal of biomechanics. 2013;46(5):905-11.

3. Buurke TJ, Lamoth CJ, van der Woude LH, Hof AL, den Otter R. Bilateral temporal control determines mediolateral margins of stability in symmetric and asymmetric human walking. ADAPTIVE CONTROL OF DYNAMIC BALANCE IN HUMAN WALKING. 2019;9:63.

4. Young PMM, Dingwell JB. Voluntarily changing step length or step width affects dynamic stability of human walking. Gait & posture. 2012;35(3):472-7.

5. Rankin BL, Buffo SK, Dean JC. A neuromechanical strategy for mediolateral foot placement in walking humans. Journal of neurophysiology. 2014;112(2):374-83.

---

## [Editor Report · Decision Letter 1]

29 Oct 2020

Active foot placement control ensures stable gait: Effect of constraints on foot placement and ankle moments

PONE-D-20-17797R1

Dear Dr. Bruijn,

I am pleased to inform you that your manuscript has been judged scientifically suitable for publication and will be formally accepted for publication once it meets all outstanding technical requirements (formatting, etc.). These requirements (if any) are determined by the production office and are separate from the content review provided by the academic editorial process.

Personally, I am happy to congratulate you on a very methodical and thorough study, and commend your group for pre-registration. Very nice work.

Kind regards,

J. Lucas McKay, Ph.D., M.S.C.R.

Academic Editor

PLOS ONE

---

## [Editor Report · Acceptance letter]

24 Nov 2020

PONE-D-20-17797R1 

Active foot placement control ensures stable gait: Effect of constraints on foot placement and ankle moments 

Dear Dr. Bruijn:

I'm pleased to inform you that your manuscript has been deemed suitable for publication in PLOS ONE. Congratulations! Your manuscript is now with our production department. 

Kind regards, 

on behalf of

Dr. J. Lucas McKay 

Academic Editor

PLOS ONE